# Fantastic Robustness Measures:
# The Secrets of Robust Generalization

**Hoki Kim**
Seoul National University
ghrl9613@snu.ac.kr

**Jinseong Park**
Seoul National University
jinseong@snu.ac.kr

**Yujin Choi**
Seoul National University
uznhigh@snu.ac.kr

**Jaewook Lee**
Seoul National University
jaewook@snu.ac.kr

## Abstract

Adversarial training has become the de-facto standard method for improving the robustness of models against adversarial examples. However, robust overfitting remains a significant challenge, leading to a large gap between the robustness on the training and test datasets. To understand and improve robust generalization, various measures have been developed, including margin, smoothness, and flatness-based measures. In this study, we present a large-scale analysis of robust generalization to empirically verify whether the relationship between these measures and robust generalization remains valid in diverse settings. We demonstrate when and how these measures effectively capture the robust generalization gap by comparing over 1,300 models trained on CIFAR-10 under the $L_\infty$ norm and further validate our findings through an evaluation of more than 100 models from RobustBench [12] across CIFAR-10, CIFAR-100, and ImageNet. We hope this work can help the community better understand adversarial robustness and motivate the development of more robust defense methods against adversarial attacks.

## 1 Introduction

Deep neural networks have achieved tremendous success in various domains, but their vulnerability to subtle perturbations has been revealed through the existence of adversarial examples, which are not generally perceptible to human beings [57, 20]. To obtain robustness against these adversarial examples, numerous defense methods have been proposed, and among them, adversarial training has become a common algorithm because of its effectiveness and ease of implementation [42, 71, 61]. However, researchers have recently found that adversarial training methods also suffer from the problem of overfitting [63, 52], where an adversarially trained model shows high robustness on training examples, yet significantly reduced robustness on test examples. As this robust overfitting progresses, the robust generalization gap increases, resulting in poor robustness for unseen examples.

To prevent robust overfitting and achieve high robust generalization, researchers have analyzed the properties of adversarial training and demonstrated the usefulness of some measures, such as margin-based measures, flatness-based measures, and gradient-based measures [53, 67, 66, 56]. Researchers have used them to estimate the robust generalization gap of given models and further developed new training schemes for improving its robustness [44, 66]. While these measures offer significant insights into robust generalization, we find that the evaluation of some measures is often limited due to the lack of models or training setups. These limitations can potentially lead to misleading conclusions, which may include inaccurate estimations of the robust generalization gap and misguided directions for the further development of adversarial training methods.

37th Conference on Neural Information Processing Systems (NeurIPS 2023).

Therefore, to gain a more precise understanding of when and how these measures correlate with robust generalization, we train over 1,300 models on CIFAR-10 under the $L_\infty$ norm across various training settings. We then investigate the relationships between a wide range of measures and their robust generalization gap. To further validate our findings, we also analyze over 100 models provided in RobustBench [12] across CIFAR-10, CIFAR-100, and ImageNet. Based on our large-scale study, we summarize our key findings as follows:

**Key findings.**

1. Due to the high sensitivity of the robust generalization gap to different training setups, the expectation of rank correlation across a wide range of training setups leads to high variance and may not capture the underlying trend.

2. Margin and smoothness exhibit significant negative correlations with the robust generalization gap. This suggests that, beyond a certain threshold, maximizing margin and minimizing smoothness can lead to a degradation in robust generalization performance.

3. Flatness-based measures, such as estimated sharpness, tend to exhibit poor correlations with the robust generalization gap. Rather, contrary to conventional assumptions, models with sharper minima can actually result in a lower robust generalization gap.

4. The norm of the input gradients consistently and effectively captures the robust generalization gap, even across diverse conditions, including fixed training methods and when conditioned on average cross-entropy.

To promote reproducibility and transparency in the field of deep learning, we have integrated the code used in this paper, along with pre-trained models, accessible to the public at `https://github.com/Harry24k/MAIR`. We hope that our findings and codes can help the community better understand adversarial robustness and motivate the development of more robust defense methods against adversarial attacks.

## 2 Related Work

The primary distinction between standard and adversarial training is that adversarial training aims to correctly classify not only benign examples but also adversarial examples as follows:

$$\min_{\boldsymbol{w}} \max_{\|\boldsymbol{x}^{adv} - \boldsymbol{x}\| \leq \epsilon} \mathcal{L}(f(\boldsymbol{x}^{adv}, \boldsymbol{w}), y), \tag{1}$$

where $(\boldsymbol{x}, y)$ is drawn from the training dataset $\mathcal{S}$ and $f$ represents the model with trainable parameters $\boldsymbol{w}$. Among the distance metrics $\|\cdot\|$, in this paper, we focus on robustness with respect to the $L_\infty$ norm. Note that the loss function $\mathcal{L}$ can also include $f(\boldsymbol{x}, \boldsymbol{w})$ to minimize the loss with respect to $\boldsymbol{x}$. By optimizing (1), we hope to minimize the robust error on the true distribution $\mathcal{D}$, defined as:

$$\mathcal{E}(\boldsymbol{w}; \epsilon, \mathcal{D}) = \mathbb{E}_{(\boldsymbol{x},y) \in \mathcal{D}} \left[ \max_{\|\boldsymbol{x}^{adv} - \boldsymbol{x}\| \leq \epsilon} \mathbb{1}(f(\boldsymbol{x}^{adv}, \boldsymbol{w}) \neq y) \right], \tag{2}$$

where $\mathbb{1}(\hat{y} \neq y)$ is an indicator function that outputs 0 if the prediction $\hat{y}$ is same as the true label $y$, and 1 otherwise.

The majority of researches have focused on optimizing (1) through the development of new loss functions or training attacks. For instance, vanilla adversarial training (AT) [42] minimizes the loss of adversarial examples generated by projected gradient descent (PGD) [42] with multiple iterations. Following AT, several variations, such as TRADES [71] and MART [61], have achieved significant reductions in robust errors on various datasets through the adoption of KL divergence and the regularization on probability margins, based on theoretical and empirical analyses.

However, recent works have revealed that the adversarial training framework has a challenging generalization problem, characterized by higher Rademacher complexity [68] and larger sample complexity [54]. The overfitting problem in adversarial training has been observed as a common phenomenon across various settings [52], and it can even occur in a more catastrophic manner during single-step adversarial training [63, 31]. As robust overfitting progresses, the following robust generalization gap $g(\boldsymbol{w})$ increases,

$$g(\boldsymbol{w}) = \mathcal{E}(\boldsymbol{w}; \epsilon, \mathcal{D}) - \mathcal{E}(\boldsymbol{w}; \epsilon, \mathcal{S}). \tag{3}$$

Thus, in order to minimize $\mathcal{E}(\boldsymbol{w}; \epsilon, \mathcal{D})$, we should not only focus on minimizing the training objective function $\mathcal{E}(\boldsymbol{w}; \epsilon, \mathcal{S})$ but also on reducing the robust generalization gap $g(\boldsymbol{w})$.

To gain a deeper understanding of the robust overfitting and further reduce the robust generalization gap $g(\boldsymbol{w})$, a line of work has theoretically and empirically investigated measures, such as boundary thickness [67], local Lipschitzness [66], and flatness [56]. While these studies claim that these measures are reliable indicators of the robust generalization gap, the lack of consistency in experimental settings hinders us to identify their validity in practical scenarios. Therefore, in this work, we aim to investigate the clear relationship between these measures and the robust generalization gap by conducting a comprehensive analysis using a large set of models.

## 2.1 Comparison to Jiang et al. [27]

The pioneering study [27] explored the empirical correlations between complexity measures and generalization with a primary focus on the standard training framework. Our main contribution is delving into the realm of measures within the adversarial training framework—a context having different generalization tendencies from those of the standard training framework [53, 66]. Indeed, we observe that the metric $\psi_k$ proposed in [27] has limitations in capturing the effectiveness of measures due to the high sensitivity of the robust generalization gap with respect to training setups. By introducing a new metric $\pi_k$, our work enhances the understanding of when and how robustness measures correlate with robust generalization. Furthermore, while Jiang et al. [27] employed customized parameter-efficient neural networks, we adopt widely-used model architectures such as ResNets, thereby providing insights that are not only relevant to recent research but also offer more practical implications.

## 3 Experimental Methodology

In the adversarial training framework, measures have played a crucial role by providing either theoretical upper bounds or empirical correlations with robust generalization. Previous works have leveraged these measures to propose new training schemes [61, 67, 64] and suggested directions to achieve high robustness [66, 56]. However, we discover that certain limitations and confusions exist when extending the findings of prior works to practical scenarios due to the use of a restricted set of models and training setups [67, 64, 66]. In order to gain a comprehensive understanding of the true efficacy of these measures, it is crucial to validate whether the effectiveness of measures remains valid in practical settings.

To address these challenges, our work aims to provide a comprehensive and accurate assessment of the effectiveness of measures for robust generalization in practical settings. Our objective is to address the fundamental question:

*Do measures remain effective in correlating with robust generalization in practical settings?*
*If so, how and when are measures correlated with robust generalization?*

To this end, in Sections 3.1 and 3.2, we carefully construct the training space that considers practical scenarios within the adversarial training framework and gather a wide range of measures from previous works. In Section 3.3, we define the evaluation metrics and introduce specific variations to accurately capture the correlation between measures and robust generalization in practical settings.

## 3.1 Training Space

In the realm of adversarial training, various training procedures have been extensively explored to enhance adversarial robustness based on the development of AT, TRADES, and MART. Recently, to resolve the issue of robust overfitting, researchers have begun combining additional techniques, including commonly employed in the standard training framework, such as early-stopping [52], using additional data [9, 22], manipulating training tricks [48, 11], and adopting sharpness-aware minimization [64]. By integrating these techniques into AT, TRADES, and MART, high adversarial robustness have been achieved, outperforming other variants of adversarial training methods [21].

Based on these prior works, to mimic practical scenarios, we have carefully selected eight training parameters widely used for improving robust generalization: (1) Model architecture {ResNet18 [23],

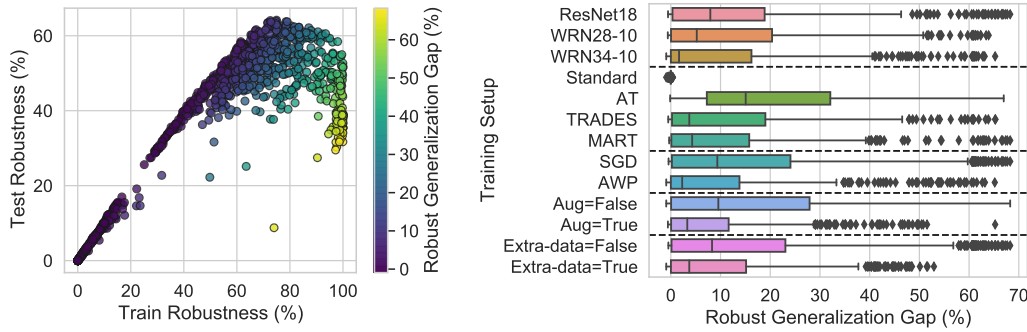

Figure 1: (Left) Scatter plot of train robustness and test robustness. Bright color corresponds to high robust generalization gap, i.e., poor generalization. (Right) Boxplot of robust generalization gap for some training setups. All adversarial examples during training and testing are generated by PGD10 on CIFAR-10.

WRN28-10 [70], WRN34-10 [70]}, (2) Training methods {Standard, AT [42], TRADES [71], MART [61]}, (3) Inner maximization steps {1, 10}, (4) Optimizer {SGD, AWP [64]}, (5) Batch-size {32, 64, 128, 256}, (6) Data augmentation {No Augmentation, Use crop and flip}, (7) Extra-data {No extra data, Use extra data [9]}, and (8) Early-stopping {No early-stopping, Use early-stopping}. Additional training details can be found in Appendix C, providing a comprehensive overview of the training procedures.

In total, 1,344 models were trained using the CIFAR-10 dataset with $\epsilon = 8/255$. Given these models, we evaluate their train/test robustness against PGD with 10 iterations (denoted as PGD10). While we acknowledge the existence of stronger adversarial attacks, such as AutoAttack [11], we primarily use PGD10 due to its prevalent use in adversarial training and the high computational demands of AutoAttack. Additionally, considering the usage of PGD in calculating specific measures, such as `boundary_thickness` and `local_lip`, ensures consistency in our analysis. For a detailed discussion, please refer to the Appendix A.4.

The statistics of the trained models are summarized in Figure 1. In the left plot, we can observe that the selected range of training parameters generates a diverse set of models, exhibiting robust generalization gaps ranging from 0% to 70%. Notably, certain models achieve 100% robustness on training data against PGD10, but their maximum robustness on the test set is only 65%, highlighting the importance of robust generalization gap. The right plot is a boxplot shows the distribution of robust generalization gaps for some training setups. As described in prior works [9, 52, 64], each training setup has a significant impact on robust generalization.

## 3.2 Measures

Beyond the measures proposed under the adversarial training frameworks [67, 66], previous works [27, 17] have demonstrated that certain measures can effectively capture the generalization gap under the standard training framework. Therefore, in this paper, we have gathered diverse measures from both the standard and adversarial training frameworks and categorized them into five different types based on their origins and formulas: (i) weight-norm (7, 8, 9, 10), (ii) margin (12, 13, 14, 15), (iii) smoothness (16, 17), (iv) flatness (18, 19, 20, 21), and (v) gradient-norm (22, 23). We denote the chosen measures in teletype font (e.g., `path_norm`). While we briefly introduce the concepts of measures in each paragraph in Section 4, we refer the readers to Appendix B for the details of measures including their mathematical definitions due to the page limit.

Given these measures, we calculate their value on whole training examples for each trained model. This choice aligns with prior works, which argue that the most direct approach for studying generalization is to prove a generalization bound that can be calculated on the training set [27] and offer a caution against the oversimplified notion that maximizing (or minimizing) a measure value inherently leads to improved generalization [5].

## 3.3 Evaluation Metrics

To uncover the relationship between measures and robust generalization performance, we adopt the Kendall rank correlation coefficient following prior works [27, 36]. We begin by defining a search space $\mathbf{\Theta} = \Theta_1 \times \Theta_2 \times \cdots \times \Theta_n$. Each $\Theta_i$ corresponds to a search space for each training parameter defined in Section 3.1. Given the search space $\mathbf{\Theta}$, we obtain the trained models $f_{\boldsymbol{\theta}}(\boldsymbol{w})$ for $\boldsymbol{\theta} \in \mathbf{\Theta}$. For each model $f_{\boldsymbol{\theta}}(\boldsymbol{w})$, we measure the robust generalization gap $g(f_{\boldsymbol{\theta}}(\boldsymbol{w}))$ and the corresponding measure value $\mu(f_{\boldsymbol{\theta}}(\boldsymbol{w}))$. For simplicity, we denote $g(\boldsymbol{\theta}) := g(f_{\boldsymbol{\theta}}(\boldsymbol{w}))$ and $\mu(\boldsymbol{\theta}) := \mu(f_{\boldsymbol{\theta}}(\boldsymbol{w}))$. We then calculate Kendall's rank coefficient [29] as follows:

$$\tau(\mathbf{\Theta}) = \frac{1}{|\mathbf{\Theta}|(|\mathbf{\Theta}| - 1)} \sum_{\boldsymbol{\theta} \in \mathbf{\Theta}} \sum_{\boldsymbol{\theta}' \in \mathbf{\Theta}, \boldsymbol{\theta} \neq \boldsymbol{\theta}'} \mathrm{sgn}(g(\boldsymbol{\theta}) - g(\boldsymbol{\theta}')) \cdot \mathrm{sgn}(\mu(\boldsymbol{\theta}) - \mu(\boldsymbol{\theta}')), \qquad (4)$$

where $|\mathbf{\Theta}|$ is the number of elements in $\mathbf{\Theta}$, and $\mathrm{sgn}(\cdot)$ is a sign function. The value of $\tau$ becomes $1$ when the pairs have the same rankings and $-1$ when the pairs have reversed order rankings. Therefore, a higher value of $\tau$ implies that as the value of a measure $\mu$ increases, the robust generalization gap $g$ also increases.

As noted by [27], the measure may strongly correlate with the robust generalization gap only when a specific training setting is varied. Therefore, Jiang et al. [27] introduced the following metric:

$$\psi_k(\mathbf{\Theta}) = \mathbf{E}_{\theta_1, \cdots, \theta_{k-1}, \theta_{k+1}, \cdots, \theta_n} \left[ \tau(\{\boldsymbol{\theta} = (\theta_1, \cdots, \theta_n), \theta_k \in \Theta_k\}) \right], \qquad (5)$$

which captures the robust generalization gap when only the hyper-parameter $\Theta_k$ changes. However, we demonstrate that $\psi_k$ may not perform well in cases where Simpson's paradox exists. Simpson's paradox refers to a situation where each of the individual groups exhibits a specific trend, but it disappears (or reverses) when the groups are combined. Thus, when a parameter $\Theta_{i \neq k}$ heavily affects the robust generalization gap, $\psi_k$ becomes not effective as it captures the overall trends by taking expectation across all parameters including $\Theta_{i \neq k}$. In fact, within the adversarial training framework, the inner $\tau$ in (5) shows extremely high variance due to the high sensitivity of the robust generalization gap with respect to training setups, which will be discussed in Table 1.

Therefore, we propose a metric for capturing the robust generalization performance by fixing the hyper-parameter $\Theta_k$ as follows:

$$\pi_k(\mathbf{\Theta}) = \mathbf{E}_{\theta_k} \left[ \tau(\{\boldsymbol{\theta} = (\theta_1, \cdots, \theta_n), \theta_i \in \Theta_i \, \forall i \neq k\}) \right]. \qquad (6)$$

Here, $\pi_k(\mathbf{\Theta})$ represents the effectiveness of a measure $\mu$ within a specific fixed training setup. This enables us to discover that some measures only work for specific training settings, e.g., uncovering the strong correlation between `boundary_thickness` and the robust generalization gap when AT is used as the training method. Furthermore, for measures exhibiting meaningful value of $\pi_k$, we additionally provide their scatter plots to mitigate the limitations of a correlation analysis.

# 4 Experimental Results

Based on the measures and trained models defined in Section 3, we calculate the measures for each model and their robust generalization gap. Notably, in the realm of adversarial training, models encounter two types of examples: benign examples and adversarial examples. Therefore, to gain a deeper understanding of the relationship between robust generalization and measures, we also calculate the values of example-dependent measures for both benign examples and PGD10 examples.

Table 1 summarizes the results of $\psi_k$ for each measure. Further, in order to consider the distributional correlation [17] and quantify the precision of $\psi_k$, we also report the corresponding standard deviation of the inner $\tau$ in Eq. (5). A higher value of $\psi_k$ indicates a stronger positive rank correlation, implying that as the measure value increases, the robust generalization gap increases. First of all, it is important to note that **none of the measures are perfect**. While certain measures show high $\psi_k$, all measures show high standard deviations. This observation indicates that no measure can provide a perfect estimation of the model's robust generalization gap. Moreover, with such high variances, it becomes challenging to clearly identify the underlying correlation of the measures.

To address this limitation, we conduct further analyses with our proposed metric, $\pi_k$ in Eq. (6). The benefit of $\pi_k$ is that it reveals the potential hidden relationship between the measures and the robust generalization gap by fixing training settings, which $\psi_k$ cannot captures. Thus, from now on, we will report $\pi_k$ of each measure and provide correlation analyses with the robust generalization gap, extending the connections beyond those presented in Table 1.

Table 1: Numerical results of $\psi_k$ for each measure, along with its corresponding standard deviation. The total $\tau$ indicates the Kendall's rank coefficient for the entire pairs $(g, \mu)$. *(PGD) indicates the same measure calculated on PGD10 examples for example-based measures.

| | Model Architecture | Training Methods | Steps | Optimizer | Batch-size | Aug | Extra-data | Early Stopping | Total $\tau$ |
|---|---|---|---|---|---|---|---|---|---|
| num_params (7) | 0.05±0.47 | - | - | - | - | - | - | - | -0.03 |
| path_norm (8) | 0.22±0.59 | 0.47±0.39 | 0.38±0.92 | 0.41±0.69 | 0.20±0.47 | 0.32±0.73 | 0.29±0.72 | 0.28±0.63 | 0.35 |
| log_prod_of_spec (9) | -0.13±0.49 | 0.15±0.40 | 0.11±0.99 | 0.16±0.75 | 0.13±0.47 | 0.02±0.73 | 0.11±0.74 | 0.17±0.66 | -0.13 |
| log_prod_of_fro (10) | 0.07±0.44 | 0.48±0.39 | 0.62±0.79 | 0.56±0.63 | 0.45±0.46 | 0.38±0.65 | 0.35±0.70 | 0.37±0.69 | 0.18 |
| euclid_init_norm (11) | 0.07±0.47 | 0.32±0.34 | 0.00±1.00 | 0.08±0.74 | -0.02±0.45 | 0.20±0.69 | 0.16±0.68 | -0.06±0.70 | 0.18 |
| average_ce (12) | -0.16±0.62 | -0.25±0.52 | -0.10±1.00 | -0.26±0.74 | -0.26±0.56 | -0.20±0.73 | -0.24±0.72 | -0.21±0.65 | -0.23 |
| inverse_margin (13) | 0.19±0.57 | 0.25±0.37 | 0.45±0.89 | 0.08±0.73 | 0.11±0.49 | 0.15±0.70 | 0.08±0.71 | 0.03±0.72 | 0.07 |
| prob_margin (14) | 0.19±0.61 | 0.23±0.52 | 0.09±1.00 | 0.26±0.74 | 0.30±0.53 | 0.25±0.72 | 0.30±0.70 | 0.20±0.65 | 0.24 |
| boundary_thickness (15) | -0.06±0.61 | 0.05±0.45 | 0.07±1.00 | -0.15±0.72 | -0.17±0.53 | -0.08±0.71 | -0.16±0.70 | -0.11±0.66 | -0.02 |
| kl_divergence (16) | -0.42±0.53 | -0.36±0.36 | -0.71±0.71 | -0.26±0.70 | -0.36±0.46 | -0.51±0.66 | -0.38±0.69 | -0.45±0.74 | -0.45 |
| local_lip (17) | -0.25±0.56 | -0.20±0.41 | -0.48±0.88 | -0.13±0.71 | -0.17±0.48 | -0.38±0.69 | -0.20±0.69 | -0.15±0.72 | -0.23 |
| pacbayes_flat (18) | 0.22±0.60 | 0.20±0.42 | 0.33±0.94 | 0.17±0.75 | 0.17±0.53 | 0.05±0.71 | 0.06±0.72 | 0.20±0.69 | 0.05 |
| estimated_sharpness (19) | 0.07±0.60 | 0.07±0.41 | 0.19±0.98 | 0.15±0.80 | 0.05±0.56 | -0.11±0.71 | -0.03±0.74 | -0.03±0.72 | -0.07 |
| estimated_inv_sharpness (20) | 0.16±0.60 | 0.20±0.35 | 0.42±0.91 | 0.27±0.77 | 0.14±0.55 | 0.00±0.71 | 0.07±0.73 | 0.06±0.78 | 0.04 |
| average_flat (21) | -0.30±0.53 | -0.31±0.41 | -0.58±0.82 | -0.21±0.68 | -0.22±0.44 | -0.43±0.66 | -0.32±0.68 | -0.30±0.71 | -0.36 |
| x_grad_norm (22) | -0.36±0.56 | -0.29±0.36 | -0.62±0.78 | -0.31±0.74 | -0.40±0.45 | -0.51±0.65 | -0.45±0.70 | -0.50±0.69 | -0.42 |
| w_grad_norm (23) | 0.17±0.56 | 0.23±0.36 | 0.50±0.86 | 0.31±0.74 | 0.16±0.52 | 0.03±0.70 | 0.10±0.72 | 0.10±0.78 | 0.07 |
| average_ce(PGD) (12) | -0.59±0.52 | -0.85±0.15 | -0.86±0.51 | -0.59±0.69 | -0.63±0.48 | -0.61±0.66 | -0.59±0.68 | -0.58±0.72 | -0.78 |
| inverse_margin(PGD) (13) | 0.56±0.49 | 0.40±0.31 | 0.75±0.66 | 0.33±0.67 | 0.44±0.43 | 0.49±0.63 | 0.39±0.67 | 0.27±0.76 | 0.35 |
| prob_margin(PGD) (14) | 0.59±0.53 | 0.80±0.14 | 0.84±0.55 | 0.61±0.69 | 0.64±0.45 | 0.62±0.65 | 0.61±0.68 | 0.59±0.71 | 0.79 |
| pacbayes_flat(PGD) (18) | 0.55±0.47 | 0.64±0.21 | 0.82±0.58 | 0.61±0.47 | 0.62±0.29 | 0.57±0.44 | 0.50±0.46 | 0.68±0.57 | 0.59 |
| estimated_sharpness(PGD) (19) | -0.05±0.64 | -0.19±0.48 | -0.10±0.99 | 0.08±0.77 | -0.03±0.55 | -0.15±0.74 | -0.12±0.75 | -0.00±0.67 | -0.22 |
| estimated_inv_sharpness(PGD) (20) | -0.08±0.65 | -0.20±0.52 | -0.13±0.99 | 0.08±0.78 | -0.06±0.57 | -0.21±0.74 | -0.12±0.75 | -0.06±0.66 | -0.25 |
| x_grad_norm(PGD) (22) | -0.23±0.54 | -0.21±0.38 | -0.53±0.85 | -0.19±0.70 | -0.26±0.45 | -0.38±0.68 | -0.29±0.70 | -0.33±0.73 | -0.34 |
| w_grad_norm(PGD) (23) | -0.14±0.63 | -0.28±0.46 | -0.19±0.98 | 0.01±0.78 | -0.09±0.57 | -0.19±0.76 | -0.16±0.76 | -0.13±0.68 | -0.32 |

Table 2: (Norm-based measures) Numerical results of $\pi_k$ and its corresponding standard deviation.

| | Model Architecture | Training Methods | Steps | Optimizer | Batch-size | Aug | Extra-data | Early Stopping |
|---|---|---|---|---|---|---|---|---|
| num_params (7) | - | -0.02±0.04 | -0.01±0.08 | -0.03±0.01 | -0.02±0.03 | -0.02±0.02 | -0.03±0.00 | -0.02±0.01 |
| path_norm (8) | 0.35±0.05 | **0.27±0.13** | **0.24±0.26** | **0.34±0.03** | **0.37±0.03** | **0.35±0.05** | **0.35±0.03** | **0.46±0.14** |
| log_prod_of_spec (9) | 0.03±0.13 | -0.07±0.07 | -0.00±0.01 | -0.12±0.03 | -0.11±0.05 | -0.13±0.01 | -0.13±0.01 | -0.13±0.06 |
| log_prod_of_fro (10) | **0.37±0.03** | 0.15±0.11 | 0.09±0.10 | 0.18±0.01 | 0.18±0.04 | 0.19±0.00 | 0.19±0.02 | 0.19±0.04 |
| euclid_init_norm (11) | 0.32±0.03 | 0.06±0.03 | 0.09±0.10 | 0.17±0.00 | 0.20±0.04 | 0.17±0.01 | 0.17±0.03 | 0.20±0.11 |

**Norm-based measures requires fixed model architecture.** In many prior works, researchers have demonstrated the effectiveness of norm-based measures in estimating the generalization gap [27, 17]. Among them, the weight norm-based measures, e.g., the product of Frobenius norm (log_prod_of_fro) [46], the product of spectral norm (log_prod_of_spec) [7], and the distance to the initial weight (euclid_init_norm) [27, 40], are built on theoretical frameworks such as PAC-Bayes [43, 47, 38]. path_norm is also often used to estimate the complexity of a neural network, which calculates the sum of outputs for all-ones input after squaring all parameters [27].

In Table 1, the most of norm-based measures exhibit a low correlation with the robust generalization gap for the metric $\psi_k$. However, by using the proposed metric $\pi_k$ in Table 2, we discover that log_prod_of_fro and euclid_init_norm exhibit strong correlations with low standard deviation when the model architecture is fixed. Intuitively, when the model architecture varies, the number of parameters and their corresponding values exhibit different ranges. Indeed, the range of log_prod_of_fro roughly shows $[50, 100]$ for ResNet18, but shows $[140, 200]$ for WRN28-10. Thus, comparing models with different architectures degrades the precision of weight norm-based measures. Under the fixation of the model architecture, **log_prod_of_fro is positively correlated with the robust generalization gap**, which consistents to the prior theoretical observations in PAC-Bayesian framework [50] or Lipschitz analysis [58]. For log_prod_of_spec, we do not observe a strong correlation under any conditions.

Furthermore, we observe that path_norm shows some extent of correlation for all $\psi_k$ and $\pi_k$. Upon conducting a more in-depth analysis, we find that **the log of path_norm yields an almost linear relationship with the robust generalization gap** when conditioned with average_ce(PGD), resulting the total $\tau = 0.68$ (detailed in Appendix A.2). This finding suggests that path_norm can be effectively utilized for estimating the robust generalization gap, with consistent to the prior works under the standard training framework [27, 17].

Table 3: (Margin-based measures) Numerical results of $\pi_k$ and its corresponding standard deviation.

| | Model Architecture | Training Methods | Steps | Optimizer | Batch-size | Aug | Extra-data | Early Stopping |
|---|---|---|---|---|---|---|---|---|
| `average_ce` (12) | -0.24±0.03 | -0.28±0.17 | -0.30±0.27 | -0.23±0.00 | -0.23±0.04 | -0.24±0.06 | -0.23±0.06 | -0.34±0.31 |
| `inverse_margin` (13) | 0.07±0.05 | -0.05±0.23 | -0.18±0.32 | 0.09±0.06 | 0.08±0.09 | 0.08±0.13 | 0.09±0.12 | 0.09±0.12 |
| `prob_margin` (14) | 0.24±0.02 | 0.28±0.16 | 0.31±0.27 | 0.23±0.01 | 0.23±0.04 | 0.24±0.06 | 0.23±0.06 | 0.34±0.31 |
| `boundary_thickness` (15) | -0.02±0.01 | -0.17±0.15 | -0.19±0.23 | -0.02±0.01 | -0.02±0.02 | -0.04±0.06 | -0.02±0.05 | -0.04±0.27 |
| `average_ce`(PGD) (12) | -0.78±0.01 | -0.58±0.39 | -0.47±0.40 | **-0.79±0.04** | **-0.79±0.01** | **-0.80±0.03** | **-0.79±0.00** | **-0.78±0.01** |
| `inverse_margin`(PGD) (13) | 0.34±0.02 | 0.26±0.22 | 0.05±0.19 | 0.37±0.03 | 0.35±0.03 | 0.34±0.05 | 0.36±0.13 | 0.38±0.08 |
| `prob_margin`(PGD) (14) | **0.79±0.01** | **0.59±0.38** | **0.48±0.39** | **0.79±0.04** | **0.79±0.01** | **0.80±0.03** | **0.79±0.00** | **0.78±0.02** |

**Maximizing margin beyond a certain point harms robust generalization.** Traditionally, the maximizing margin is considered as an ultimate goal in the adversarial training framework [42, 61]. Indeed, average cross-entropy loss [42] and margin-related losses [61] are frequently used in adversarial training methods. However, in both Table 1 and Table 3, `average_ce`(PGD) exhibit a high negative correlation with the robust generalization gap across all variations of training parameters. Similarly, $\pi_k$ and the total $\tau$ of `prob_margin`(PGD) are extremely high (0.79). This suggests that **lower cross-entropy on PGD examples (and higher margin in the probability space) leads to worse robust generalization.** Indeed, Fig. 2 shows a clear negative correlation between `average_ce`(PGD) and the robust generalization gap. Notably, high test robustness is observed within the range of `average_ce`(PGD) $\in [0.5, 1.0]$, indicating that minimizing `average_ce`(PGD) beyond a certain point may harm the generalization performance as Ishida et al. [25] observed in the standard training frameworks.

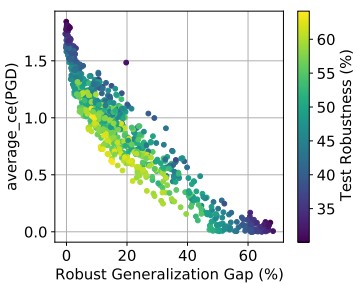

Figure 2: Scatter plot for `average_ce`(PGD) and the gap. Bright color indicates a higher test robustness. For better visualization, we cutoff `average_ce`(PGD) $> 2$.

Given this observation, we argue that the margin maximization in adversarial training methods should be carefully revisited. Recent studies have highlighted that maximizing the margin might not necessarily be the optimal objective in adversarial training due to intricate gradient flow dynamics [59] and the non-cognitive concept of using predicted probabilities [1]. Additionally, in recent work [32], despite the similar robustness of TRADES and AT, their margin distributions on benign and adversarial examples are extremely different. This implies that the margin cannot be the sole determinant of adversarial robustness. Considering these findings and other recent studies [41, 60], the margin maximization should be accompanied by a consideration of other factors such as weight regularization or gradient information.

The cross entropy and margin on benign examples, denoted as `average_ce` and `prob_margin`, also show some degree of correlation with the generalization gap. This correlation becomes particularly significant when using early stopping, where their correlations reach up to 0.65. Note that `inverse_margin` behaves differently because it uses the 10th-percentile of margins over the training dataset rather than the expectation.

**Boundary thickness works well for models trained by AT.** Yang et al. [67] introduced the concept of boundary thickness, which is an extended version of margin based on adversarial examples. They argue that a thin decision boundary leads to both poor adversarial robustness and the gap. Therefore, `boundary_thickness` should be negatively correlated with the robust generalization gap. However, as shown in Table 3, it does not correlate well with the robust generalization gap. The main difference between our experiments and those in [67] is that we also considered TRADES and MART, whereas Yang et al. [67] sorely compared models trained with AT. Thus, in Fig. 3, we plot the inner $\tau$ in $\pi_k$ for each training method. It is evident that the boundary thickness demonstrates a strong correlation with the robust generalization gap when the training method is fixed to AT. This suggests that `boundary_thickness` **is more effective for comparing models trained with AT**. Furthermore, in Appendix A.2, we also discover that `boundary_thickness` becomes more highly correlated with the robust generalization gap when the models are conditioned on `average_ce`(PGD). Thus, when using `boundary_thickness` as the sole determinant of robust generalization, we should carefully revisit the choice of training methods and the robust accuracy of models on train datasets.

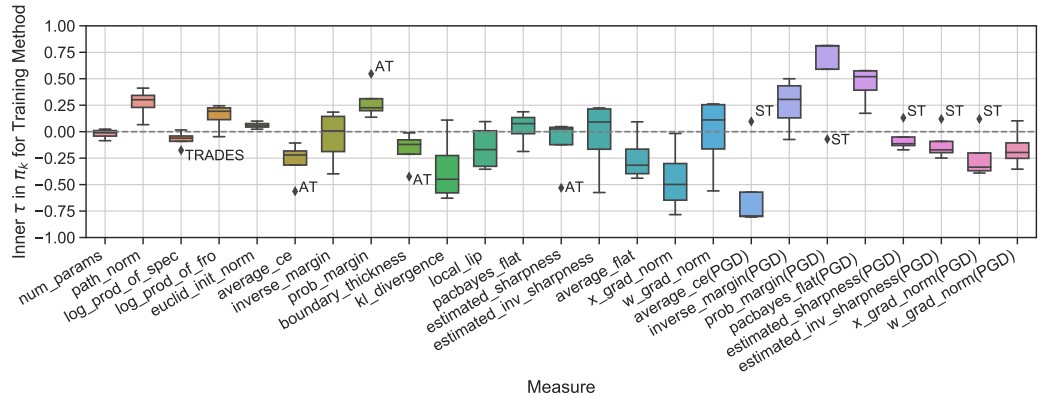

Figure 3: Box plot of the inner $\tau$ in $\pi_k$ Eq. (6), where $\Theta_k$ is training method. The text corresponds to the method for each outlier, e.g., `boundary_thickness` performs well when AT is used.

Table 4: (Smoothness-based and Flatness-based measures) Numerical results of $\pi_k$ and its corresponding standard deviation.

| | Model Architecture | Training Methods | Steps | Optimizer | Batch-size | Aug | Extra-data | Early Stopping |
|---|---|---|---|---|---|---|---|---|
| `kl_divergence` (16) | -0.45±0.01 | -0.35±0.29 | -0.15±0.19 | -0.46±0.10 | -0.46±0.08 | -0.43±0.12 | -0.45±0.02 | -0.37±0.29 |
| `local_lip` (17) | -0.24±0.06 | -0.15±0.19 | -0.01±0.17 | -0.24±0.10 | -0.25±0.07 | -0.20±0.05 | -0.23±0.02 | -0.21±0.30 |
| `pacbayes_flat` (18) | 0.04±0.09 | 0.04±0.14 | -0.06±0.14 | 0.06±0.07 | 0.04±0.08 | 0.10±0.12 | 0.08±0.10 | 0.07±0.16 |
| `estimated_sharpness` (19) | -0.07±0.02 | -0.11±0.24 | -0.22±0.18 | -0.06±0.17 | -0.08±0.10 | -0.02±0.13 | -0.04±0.07 | -0.04±0.16 |
| `estimated_inv_sharpness` (20) | 0.04±0.02 | -0.04±0.32 | -0.19±0.14 | 0.05±0.12 | 0.04±0.06 | 0.10±0.13 | 0.07±0.10 | 0.03±0.12 |
| `average_flat` (21) | -0.36±0.01 | -0.24±0.21 | -0.10±0.15 | -0.38±0.07 | -0.37±0.06 | -0.32±0.13 | -0.35±0.00 | -0.33±0.25 |
| `pacbayes_flat(PGD)` (18) | **0.59±0.04** | **0.45±0.16** | **0.30±0.21** | **0.60±0.00** | **0.59±0.02** | **0.62±0.02** | **0.63±0.03** | **0.50±0.07** |
| `estimated_sharpness(PGD)` (19) | -0.22±0.01 | -0.07±0.12 | -0.06±0.14 | -0.22±0.10 | -0.23±0.07 | -0.17±0.07 | -0.19±0.06 | -0.21±0.32 |
| `estimated_inv_sharpness(PGD)` (20) | -0.24±0.02 | -0.12±0.14 | -0.10±0.17 | -0.26±0.10 | -0.26±0.06 | -0.20±0.08 | -0.23±0.06 | -0.23±0.35 |

**Smoothness does not guarantee low robust generalization gap.** In the pursuit of achieving adversarial robustness, the smoothness between benign and adversarial examples is often considered as an indicative measure. For instance, TRADES [71] minimizes the `kl_divergence` between benign and adversarial logits. However, `kl_divergence` shows a negative correlation for both Table 1 and Table 4. This implies that, similar to `average_ce(PGD)`, `kl_divergence` cannot serve as an indicator for robust generalization.

While Xu et al. [65] demonstrated that imposing local Lipschitzness (`local_lip`) leads to better generalization in linear classification, recent research [66] argued an opposing perspective, suggesting that within neural networks, local Lipschitzness might hurt robust generalization. However, this conclusion was built on fewer than 20 models and evaluated solely on test examples. In our large experiment, we cannot observe that local Lipschitzness itself negatively affects robust generalization. Rather, it is more efficient in predicting robust accuracy (detailed in Appendix A.1). These findings are consistent with [45, 39], which highlighted the importance of model architecture or weight norms when evaluating models with local Lipshitzness.

**Flatness-based measures are not correlated well with the robust generalization gap.** Flatness-based measures have recently regarded as powerful indicators of generalization performance in both standard and adversarial training frameworks [18, 64]. This includes the maximum perturbation size in the weight space that do not dramatically changes the accuracy (`pacbayes_flat`) [43, 27], the loss increment by adversarial weight perturbation (`estimated_sharpness`) [18], and its scale-invariant version (`estimated_inv_sharpness`) [36]. However, our analysis reveals that **flatness-based measures tend to exhibit poor correlations with the robust generalization gap**. In both Table 1 and Table 4, the majority of flatness-based measures exhibit near-zero correlations or even negative values. Only `pacbayes_flat(PGD)` demonstrates a strong correlation with robust generalization because it effectively distinguishes between robust and non-robust models (refer to Appendix A.2).

Recently, Stutz et al. [56] demonstrated the importance of early stopping in the analysis of flatness. Similarly, we observe that, when early stopping is employed, the correlation of `estimated_sharpness` approaches zero. However, without early-stopping, we discover

Table 5: (Gradient-based measures) Numerical results of $\pi_k$ and its corresponding standard deviation.

| | Model Architecture | Training Methods | Steps | Optimizer | Batch-size | Aug | Extra-data | Early Stopping |
|---|---|---|---|---|---|---|---|---|
| `x_grad_norm` (22) | **-0.42±0.03** | **-0.45±0.28** | **-0.31±0.23** | **-0.43±0.09** | **-0.42±0.02** | **-0.40±0.02** | **-0.41±0.05** | **-0.38±0.07** |
| `w_grad_norm` (23) | 0.07±0.04 | -0.02±0.33 | -0.17±0.18 | 0.08±0.13 | 0.07±0.08 | 0.11±0.13 | 0.09±0.09 | 0.05±0.08 |
| `x_grad_norm(PGD)` (22) | -0.34±0.02 | -0.24±0.21 | -0.09±0.15 | -0.36±0.05 | -0.34±0.09 | -0.32±0.12 | -0.34±0.02 | -0.25±0.25 |
| `w_grad_norm(PGD)` (23) | -0.32±0.03 | -0.16±0.17 | -0.15±0.18 | -0.32±0.11 | -0.33±0.09 | -0.28±0.08 | -0.31±0.04 | -0.28±0.31 |

Table 6: Numerical results of $\pi_k$ for each measure when $\Theta_k$ is given by `average_ce(PGD)` values with a bin size of 0.1, along with its corresponding standard deviation.

| Measures | $\pi_k$ | Measures | $\pi_k$ |
|---|---|---|---|
| `num_params` (7) | -0.23±0.17 | `estimated_sharpness` (19) | -0.12±0.19 |
| `path_norm` (8) | 0.25±0.14 | `estimated_inv_sharpness` (20) | -0.13±0.17 |
| `log_prod_of_spec` (9) | 0.10±0.13 | `average_flat` (21) | -0.12±0.17 |
| `log_prod_of_fro` (10) | 0.11±0.10 | `x_grad_norm` (22) | **-0.35±0.16** |
| `euclid_init_norm` (11) | -0.11±0.14 | `w_grad_norm` (23) | -0.06±0.17 |
| `average_ce` (12) | 0.03±0.19 | `inverse_margin(PGD)` (13) | 0.05±0.19 |
| `inverse_margin` (13) | -0.09±0.19 | `prob_margin(PGD)` (14) | 0.13±0.16 |
| `prob_margin` (14) | -0.02±0.16 | `pacbayes_flat(PGD)` (18) | -0.19±0.20 |
| `boundary_thickness` (15) | 0.06±0.19 | `estimated_sharpness(PGD)` (19) | 0.05±0.19 |
| `kl_divergence` (16) | -0.09±0.14 | `estimated_inv_sharpness(PGD)` (20) | 0.01±0.19 |
| `local_lip` (17) | -0.11±0.16 | `x_grad_norm(PGD)` (22) | -0.12±0.14 |
| `pacbayes_flat` (18) | -0.25±0.21 | `w_grad_norm(PGD)` (23) | 0.13±0.16 |

that `estimated_sharpness` exhibits a significant negative correlation for models have low `average_ce(PGD)` $\leq$ 1.5. As shown in Fig. 4, models with low `estimated_sharpness` show high robust generalization gaps. This finding aligns with the concurrent work of [5], which demonstrates that flatter solutions generalize worse on out-of-distribution data. The additional results can be found in Appendix A.6.

In the case of `average_flat` [56], which is calculated with random weight perturbations and their worst-case losses, it demonstrates some degree of correlation. However, it is more efficient in predicting robust accuracy rather than the gap (refer to Appendix A.1). This result suggests that, as the concurrent work [5] observed in the standard training framework, flatness measures may not serve as reliable indicators of correlation in the adversarial training framework even they can be effectively used to achieving better performance.

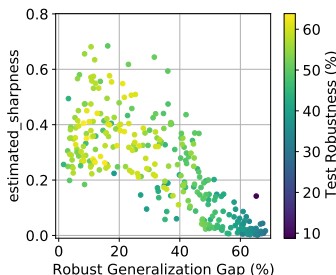

Figure 4: Scatter plot for the robust generalization gap and `estimated_sharpness`. Conditioned on `average_ce(PGD)` $\leq$ 1.5 without early-stopping.

**The norm of gradient of inputs robustly captures the gap even for models with similar cross-entropy losses.** Although some prior works [53, 24] have argued that regularizing the input gradient norm might improve adversarial robustness, we observe that this cannot be argued as lower input gradient norm is better. Table 5 summarizes $\pi_k$ of the gradient norm of input (`x_grad_norm`) and the gradient norm of weight (`w_grad_norm`). Among these, `x_grad_norm` show a strong correlation with the robust generalization gap. The negative correlation of `x_grad_norm` indicates that models with a larger input gradient norm are more likely to show lower robust generalization gap.

Furthermore, even when comparing models having similar `average_ce(PGD)`, `x_grad_norm` is the most robust indicator of the robust generalization gap. Previous works in the standard training framework [27, 17] have argued that the cross-entropy loss is strongly correlated with the robust generalization gap, and thus, they used early stopping based on certain cross-entropy thresholds during training to remove the influence of varying cross-entropy loss. However, within the adversarial training framework, employing the same early stopping based on loss becomes challenging as TRADES and MART minimize different loss functions from AT. Therefore, we categorize the trained models into groups based on `average_ce(PGD)` values using a bin size of 0.1. This grouping reduces $\pi_k$ of `average_ce(PGD)` to $-0.12$. The results are summarized in Table 6. Compared to all other measures, **`x_grad_norm` exhibits the highest rank correlation with the robust generalization gap even when conditioned on `average_ce(PGD)`.** We believe this finding highlights the importance of the norm of input gradients as a valuable regularizer for improving model robustness and its generalization in practical settings.

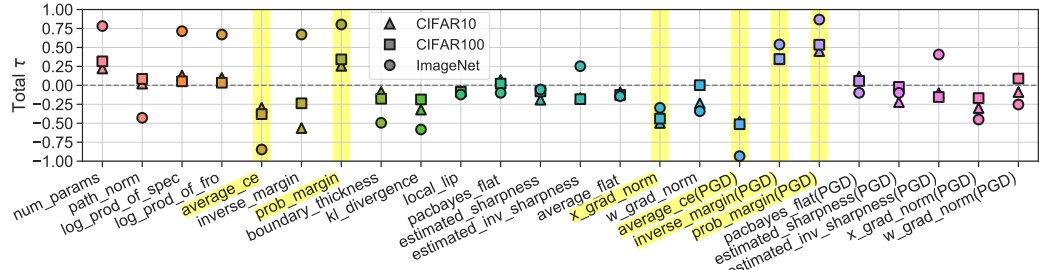

Figure 5: Experiment on RobustBench [12]. For each dataset, we plot the total $\tau$ of each measure and highlight the robust measures with $|\tau| \geq 0.25$ for all datasets with yellow background.

Table 7: Numerical results of $\pi_k$ for norm-based measures when $\Theta_k$ is the model structure along with its corresponding standard deviation. The total $\tau$ is the same as in Fig. 5. For ImageNet, $\pi_k$ is not applicable due to the limited number of pre-trained models in RobustBench [12].

| CIFAR-10 | $\pi_k$ | Total $\tau$ | CIFAR-100 | $\pi_k$ | Total $\tau$ |
|---|---|---|---|---|---|
| `path_norm` (8) | 0.24±0.26 | 0.02 | `path_norm` (8) | 0.55±0.05 | 0.09 |
| `log_prod_of_fro` (10) | 0.12±0.40 | 0.10 | `log_prod_of_fro` (10) | 0.30±0.30 | 0.17 |

## 5 Broader Impact with Benchmarks

RobustBench [12] provides a set of pre-trained models that achieve high robust accuracy across various datasets, including CIFAR-10, CIFAR-100, and ImageNet. Leveraging this benchmark, we extend our observations to diverse models including transformer-based architectures [3, 13] or trained on diffusion-generated datasets [51, 62]. As shown in Figure 5, we identify that some of our findings in Section 4 also can be applied to these models. Margin-based measures such as `average_ce`, `prob_margin`, `average_ce(PGD)`, and `prob_margin(PGD)` consistently exhibit strong correlations with the robust generalization gap. Additionally, we observe that `x_grad_norm` consistently shows reliable performance in predicting the robust generalization gap, even when applied to models in the RobustBench across various datasets.

Though a deeper analysis is limited by the absence of training setting details, such as the use of early stopping, we additionally conduct an analysis with the model architecture by analyzing the pre-trained models. Table 7 summarizes $\pi_k$ for models with the same architecture. As we demonstrated in Section 4, norm-based measures exhibit a higher correlation when comparing models with identical architectures. Notably, for CIFAR-100, we find that `path_norm` shows a strong correlation with the robust generalization gap. Regarding the low correlation and high standard deviation of norm-based measures, we hypothesize that other training settings, such as the choice of activation functions (e.g., Swish and SiLU instead of ReLU) and training datasets, may affect the values of norm-based measures. Further exploration of these aspects is left to future work, as additional research and experiments can provide a more comprehensive understanding of these relationships.

## 6 Limitations and Future Work

While our study unveils the correlation between various measures and the robust generalization gap over 1,300 models, due to our computational constraints, we focused on ResNet models, CIFAR-10, and PGD with the $L_\infty$ norm. Thus, further investigations on a broader range of hyper-parameters and the use of stronger attacks may uncover new relationships beyond our analysis. We hope that future work would address these limitations.

## 7 Conclusion

Through large-scale experiments, we verified the underlying relationships between various measures and the robust generalization gap on CIFAR-10 under the $L_\infty$ norm. Our findings offer valuable insights into robust generalization and emphasize the need for caution when making statements such as, 'model A is superior to model B because model A exhibits a better measure value than model B,' a frequently employed phrase in recent literature. We hope that our discoveries regarding diverse measures can contribute to further advancement in the field of adversarial robustness.

## Acknowledgements

This work was supported by the National Research Founda- tion of Korea (NRF) grant funded by the Korean government (MSIT) (No. 2019R1A2C2002358) and the Institute of Information & communications Technology Planning & Evaluation (IITP) grant funded by the Korea government (MSIT) (No. 2022-0-00984).

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

# A  Additional Experiments

## A.1  Estimating Test Robust Accuracy

Instead of estimating the robust generalization gap, one might expect the analysis on the relationship between test robust accuracy and the measures. In this regard, we investigate the correlation between the measures and the test robust accuracy $1 - \mathcal{E}(\boldsymbol{w}; \epsilon, \mathcal{D})$ on the test dataset $\mathcal{D}$ instead of the robust generalization gap $g(\boldsymbol{w})$.

Table 8: Numerical results of $\psi_k$ between each measure and *test robust accuracy*, along with its corresponding standard deviation.

| | Model Architecture | Training Methods | Steps | Optimizer | Batch-size | Aug | Extra-data | Early Stopping | Total $\tau$ |
|---|---|---|---|---|---|---|---|---|---|
| num_params (7) | 0.14±0.44 | - | - | - | - | - | - | - | 0.02 |
| path_norm (8) | 0.22±0.57 | 0.36±0.41 | 0.33±0.94 | 0.08±0.72 | 0.22±0.50 | 0.20±0.70 | 0.17±0.73 | -0.43±0.51 | 0.02 |
| log_prod_of_spec (9) | -0.22±0.43 | 0.06±0.42 | 0.05±1.00 | 0.04±0.73 | 0.03±0.49 | 0.07±0.73 | 0.07±0.73 | -0.16±0.68 | -0.15 |
| log_prod_of_fro (10) | 0.12±0.41 | 0.36±0.39 | 0.64±0.77 | 0.30±0.70 | 0.44±0.41 | 0.39±0.66 | 0.40±0.65 | 0.08±0.66 | 0.07 |
| euclid_init_norm (11) | 0.16±0.45 | 0.28±0.31 | -0.14±0.99 | -0.01±0.73 | -0.14±0.48 | -0.13±0.69 | -0.10±0.68 | -0.23±0.62 | 0.10 |
| average_ce (12) | -0.09±0.62 | 0.03±0.53 | 0.08±1.00 | 0.17±0.73 | -0.13±0.57 | -0.01±0.77 | 0.01±0.76 | 0.55±0.47 | 0.14 |
| inverse_margin (13) | 0.24±0.56 | 0.44±0.36 | 0.48±0.88 | 0.44±0.60 | 0.18±0.50 | 0.28±0.69 | 0.24±0.68 | 0.71±0.37 | 0.44 |
| prob_margin (14) | 0.05±0.62 | -0.03±0.54 | -0.09±1.00 | -0.16±0.74 | 0.11±0.57 | -0.01±0.77 | -0.03±0.75 | -0.56±0.46 | -0.14 |
| boundary_thickness (15) | 0.07±0.60 | 0.24±0.49 | 0.26±0.97 | 0.29±0.71 | 0.06±0.53 | 0.17±0.75 | 0.21±0.73 | 0.62±0.43 | 0.32 |
| kl_divergence (16) | -0.55±0.49 | -0.57±0.36 | -0.94±0.35 | -0.65±0.48 | -0.46±0.40 | -0.37±0.60 | -0.45±0.61 | -0.63±0.47 | -0.45 |
| local_lip (17) | -0.46±0.49 | -0.44±0.44 | -0.70±0.72 | -0.58±0.56 | -0.32±0.44 | -0.28±0.65 | -0.33±0.69 | -0.64±0.49 | -0.40 |
| pacbayes_flat (18) | 0.31±0.52 | 0.21±0.44 | 0.31±0.95 | 0.17±0.77 | 0.25±0.51 | 0.28±0.68 | 0.26±0.67 | -0.17±0.67 | 0.15 |
| estimated_sharpness (19) | 0.09±0.60 | 0.25±0.39 | 0.36±0.93 | 0.16±0.75 | 0.21±0.50 | 0.30±0.68 | 0.32±0.67 | 0.20±0.66 | 0.18 |
| estimated_inv_sharpness (20) | 0.24±0.56 | 0.39±0.33 | 0.56±0.83 | 0.29±0.74 | 0.29±0.50 | 0.38±0.65 | 0.41±0.64 | 0.51±0.58 | 0.33 |
| average_flat (21) | -0.44±0.50 | -0.60±0.41 | -0.82±0.57 | -0.64±0.51 | -0.40±0.42 | -0.30±0.62 | -0.35±0.63 | -0.66±0.46 | -0.42 |
| x_grad_norm (22) | -0.40±0.48 | -0.34±0.39 | -0.74±0.68 | -0.49±0.59 | -0.37±0.41 | -0.26±0.65 | -0.30±0.65 | -0.09±0.76 | -0.17 |
| w_grad_norm (23) | 0.20±0.52 | 0.42±0.35 | 0.66±0.75 | 0.33±0.75 | 0.29±0.50 | 0.42±0.66 | 0.42±0.65 | 0.60±0.54 | 0.37 |
| average_ce(PGD) (12) | -0.70±0.41 | -0.67±0.19 | -0.89±0.46 | -0.60±0.48 | -0.61±0.35 | -0.51±0.58 | -0.50±0.60 | -0.57±0.51 | -0.58 |
| inverse_margin(PGD) (13) | 0.58±0.47 | 0.64±0.30 | 0.86±0.50 | 0.69±0.45 | 0.51±0.43 | 0.49±0.62 | 0.59±0.56 | 0.78±0.33 | 0.61 |
| prob_margin(PGD) (14) | 0.74±0.43 | 0.73±0.18 | 0.92±0.38 | 0.66±0.43 | 0.60±0.39 | 0.54±0.58 | 0.52±0.60 | 0.58±0.50 | 0.60 |
| pacbayes_flat(PGD) (18) | 0.62±0.44 | 0.59±0.24 | 0.81±0.58 | 0.60±0.56 | 0.52±0.45 | 0.60±0.48 | 0.72±0.38 | 0.11±0.77 | 0.61 |
| estimated_sharpness(PGD) (19) | -0.17±0.65 | -0.28±0.54 | -0.25±0.97 | -0.22±0.75 | -0.06±0.57 | -0.01±0.74 | -0.01±0.72 | -0.49±0.48 | -0.22 |
| estimated_inv_sharpness(PGD) (20) | -0.20±0.65 | -0.22±0.56 | -0.22±0.98 | -0.19±0.75 | -0.05±0.56 | 0.01±0.74 | 0.04±0.71 | -0.48±0.48 | -0.21 |
| x_grad_norm(PGD) (22) | -0.37±0.51 | -0.45±0.35 | -0.77±0.64 | -0.55±0.59 | -0.32±0.44 | -0.21±0.64 | -0.29±0.66 | -0.63±0.45 | -0.38 |
| w_grad_norm(PGD) (23) | -0.30±0.61 | -0.28±0.50 | -0.28±0.96 | -0.26±0.73 | -0.09±0.57 | -0.05±0.72 | -0.05±0.73 | -0.47±0.50 | -0.28 |

Table 9: Numerical results of $\pi_k$ between each measure and *test robust accuracy*, along with its corresponding standard deviation.

| | Model Architecture | Training Methods | Steps | Optimizer | Batch-size | Aug | Extra-data | Early Stopping | Total $\tau$ |
|---|---|---|---|---|---|---|---|---|---|
| num_params (7) | - | 0.04±0.04 | 0.05±0.12 | 0.03±0.01 | 0.02±0.03 | 0.02±0.00 | 0.03±0.02 | 0.02±0.02 | 0.02 |
| path_norm (8) | 0.01±0.12 | -0.20±0.19 | -0.33±0.14 | 0.04±0.00 | 0.02±0.03 | 0.04±0.01 | 0.02±0.05 | 0.27±0.18 | 0.02 |
| log_prod_of_spec (9) | -0.06±0.12 | -0.08±0.10 | -0.00±0.09 | -0.15±0.06 | -0.14±0.05 | -0.15±0.01 | -0.15±0.03 | -0.16±0.04 | -0.15 |
| log_prod_of_fro (10) | 0.22±0.08 | 0.06±0.05 | -0.00±0.14 | 0.07±0.01 | 0.07±0.01 | 0.07±0.02 | 0.06±0.01 | 0.12±0.05 | 0.07 |
| euclid_init_norm (11) | 0.08±0.02 | -0.10±0.16 | -0.12±0.18 | 0.11±0.00 | 0.11±0.04 | 0.11±0.04 | 0.12±0.01 | 0.16±0.14 | 0.10 |
| average_ce (12) | 0.16±0.06 | 0.26±0.21 | 0.32±0.21 | 0.13±0.02 | 0.16±0.02 | 0.15±0.02 | 0.15±0.03 | -0.08±0.40 | 0.14 |
| inverse_margin (13) | 0.45±0.08 | 0.47±0.13 | 0.43±0.14 | 0.44±0.02 | 0.44±0.08 | 0.44±0.04 | 0.44±0.03 | 0.34±0.02 | 0.44 |
| prob_margin (14) | -0.16±0.05 | -0.27±0.21 | -0.32±0.22 | -0.13±0.02 | -0.15±0.02 | -0.15±0.01 | -0.15±0.03 | 0.08±0.41 | -0.14 |
| boundary_thickness (15) | 0.33±0.02 | 0.36±0.22 | 0.41±0.22 | 0.30±0.04 | 0.33±0.02 | 0.32±0.02 | 0.32±0.03 | 0.19±0.38 | 0.32 |
| kl_divergence (16) | -0.44±0.01 | -0.47±0.15 | -0.29±0.36 | -0.45±0.01 | -0.45±0.03 | -0.48±0.04 | -0.46±0.08 | -0.40±0.20 | -0.40 |
| local_lip (17) | -0.39±0.03 | -0.45±0.15 | -0.40±0.27 | -0.38±0.06 | -0.40±0.03 | -0.43±0.01 | -0.40±0.03 | -0.31±0.31 | -0.40 |
| pacbayes_flat (18) | 0.16±0.04 | 0.07±0.30 | -0.08±0.37 | 0.16±0.06 | 0.14±0.06 | 0.14±0.08 | 0.14±0.10 | 0.26±0.16 | 0.15 |
| estimated_sharpness (19) | 0.19±0.05 | 0.18±0.11 | 0.05±0.28 | 0.19±0.10 | 0.17±0.04 | 0.17±0.00 | 0.16±0.03 | 0.20±0.11 | 0.18 |
| estimated_inv_sharpness (20) | 0.35±0.06 | 0.35±0.20 | 0.20±0.15 | 0.35±0.05 | 0.33±0.03 | 0.33±0.00 | 0.32±0.01 | 0.28±0.07 | 0.33 |
| average_flat (21) | -0.42±0.02 | -0.42±0.17 | -0.31±0.38 | -0.40±0.07 | -0.42±0.02 | -0.45±0.02 | -0.43±0.03 | -0.36±0.24 | -0.42 |
| x_grad_norm (22) | -0.17±0.02 | -0.13±0.11 | 0.04±0.22 | -0.17±0.01 | -0.17±0.06 | -0.20±0.16 | -0.19±0.18 | -0.20±0.05 | -0.17 |
| w_grad_norm (23) | 0.41±0.08 | 0.39±0.22 | 0.24±0.08 | 0.38±0.06 | 0.38±0.05 | 0.36±0.02 | 0.37±0.01 | 0.29±0.01 | 0.37 |
| average_ce(PGD) (12) | -0.56±0.06 | -0.52±0.18 | -0.41±0.43 | -0.59±0.05 | -0.57±0.01 | -0.59±0.09 | -0.58±0.08 | -0.61±0.07 | -0.58 |
| inverse_margin(PGD) (13) | 0.61±0.01 | 0.64±0.04 | 0.52±0.16 | 0.61±0.04 | 0.61±0.01 | 0.63±0.01 | 0.60±0.02 | 0.56±0.15 | 0.61 |
| prob_margin(PGD) (14) | 0.58±0.06 | 0.54±0.20 | 0.45±0.44 | 0.61±0.05 | 0.59±0.02 | 0.61±0.09 | 0.60±0.09 | 0.64±0.05 | 0.60 |
| pacbayes_flat(PGD) (18) | 0.60±0.02 | 0.29±0.57 | 0.08±0.54 | 0.61±0.04 | 0.61±0.03 | 0.61±0.05 | 0.58±0.07 | 0.62±0.10 | 0.61 |
| estimated_sharpness(PGD) (19) | -0.20±0.01 | -0.21±0.23 | -0.22±0.40 | -0.20±0.08 | -0.22±0.04 | -0.24±0.02 | -0.25±0.01 | -0.15±0.32 | -0.22 |
| estimated_inv_sharpness(PGD) (20) | -0.18±0.04 | -0.23±0.24 | -0.24±0.41 | -0.19±0.06 | -0.21±0.05 | -0.23±0.04 | -0.24±0.00 | -0.13±0.34 | -0.21 |
| x_grad_norm(PGD) (22) | -0.39±0.00 | -0.41±0.13 | -0.29±0.34 | -0.37±0.04 | -0.39±0.03 | -0.41±0.01 | -0.39±0.05 | -0.27±0.16 | -0.38 |
| w_grad_norm(PGD) (23) | -0.25±0.03 | -0.28±0.24 | -0.30±0.40 | -0.27±0.07 | -0.28±0.02 | -0.31±0.01 | -0.30±0.03 | -0.21±0.28 | -0.28 |

Tables 8 and 9 present the values of $\psi_k$ and $\pi_k$ for each measure. Fig. 6 illustrates the difference in total $\tau$ when the robust generalization gap and the test robust accuracy are used as the target variable for correlation analysis. Certain measures exhibit stronger rank correlations with the test robust accuracy than the robust generalization gap. w_grad_norm, inverse_margin, and inverse_margin(PGD) exhibit the total $\tau$ values exceeding 0.4 with respect to the test robust accuracy, whereas they show

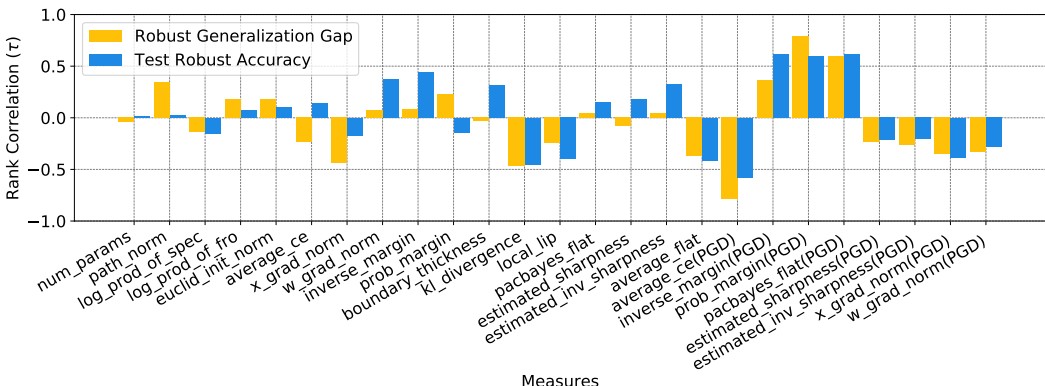

Figure 6: Comparison of the total $\tau$ when the robust generalization $g(\boldsymbol{w})$ (yellow) and the test robust accuracy $1 - \mathcal{E}(\boldsymbol{w}; \epsilon, \mathcal{D})$ (blue) are used as the target variables for correlation analysis.

near-zero total $\tau$ values for the robust generalization gap. Similarly, `local_lip` and `average_flat` show stronger correlations. Although we observe some different behavior of measures, we find that estimating the test robust accuracy can be more challenging. When we perform a linear regression analysis, predicting the test robust accuracy yields poor $R^2$ values. To ease comparison, we refer the readers detailed analysis to Table 15 in Appendix A.3.

## A.2 Focusing on Adversarially Robust models

In the main paper, we use all models regardless of their robust accuracy, to ensure the generality of our analyses. However, as we discussed in Table 4, adversarially robust and non-robust models may exhibit different behaviors with some measures. For instance, as shown in Fig. 7, non-robust models have extremely low values of `pacbayes_flat(PGD)` less 5. In contrast, robust models show higher values of `pacbayes_flat(PGD)` over 5. Thus, investigating only robust models might potentially reveal hidden behaviors of measures in predicting the robust generalization gap.

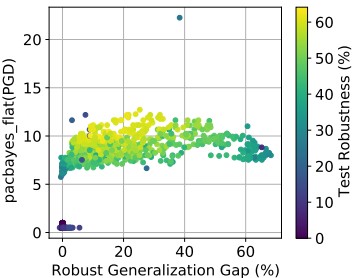

Figure 7: Scatter plot of `pacbayes_flat(PGD)`. Adversarially robust models (bright colors) and non-robust models (darker colors) show distinct range of values for `pacbayes_flat(PGD)`.

In Tables 10 and 11, we summarize $\psi_k$ and $\pi_k$ with conditioned on `average_ce(PGD)` $\leq 1.5$. To ease the comparison between the previous results, we also illustrate the total $\tau$ of those in Fig. 8. First, overall $\pi_k$ and $\psi_k$ of `path_norm` increases, and the total $\tau$ increases 0.35 to 0.63. As shown in Fig. 9a, the log of `path_norm` shows almost linear relationships between the robust generalization gap. Similarly, overall $\pi_k$ and $\psi_k$ of `prob_margin`, and the total $\tau$ decreased -0.23 to -0.62. Similar to `prob_margin`, the margin-based measures, i.e., `inverse_margin` and `average_ce`, show more negative correlations. Thus, similar to probability margins on adversarial examples, maximizing the margins on benign examples may harm to the robust generalization with high probability. Lastly, `boundary_thickness` shows strong correlation to the robust generalization gap for `average_ce(PGD)` $\leq 1.5$. As shown in Fig. 9c, high `boundary_thickness` shows low robust generalization gap. Thus, we can conclude that `average_ce(PGD)` is also an effective condition for `boundary_thickness` as well as the training method.

Notably, some flatness-based measures have a weaker correlation, `average_flat` (-0.36 to -0.15), `estimated_sharpness(PGD)` (-0.22 to -0.02), and `estimated_inv_sharpness(PGD)` (-0.25 to 0.03). Their overall values of $\psi_k$ and $\pi_k$ in Tables 10 and 11 are also close to 0, which supports our claim that flatness measures cannot serve as reliable indicators of correlation in the adversarial training framework.

Table 10: Numerical results of $\psi_k$ for each measure, along with its corresponding standard deviation. Conditioned on `average_ce(PGD)` $\leq 1.5$.

| | Model Architecture | Training Methods | Steps | Optimizer | Batch-size | Aug | Extra-data | Early Stopping | Total $\tau$ |
|---|---|---|---|---|---|---|---|---|---|
| num_params (7) | 0.23±0.64 | - | - | - | - | - | - | - | 0.04 |
| path_norm (8) | 0.43±0.54 | 0.45±0.50 | 0.54±0.84 | 0.77±0.46 | 0.32±0.52 | 0.71±0.49 | 0.71±0.51 | 0.88±0.30 | 0.63 |
| log_prod_of_spec (9) | -0.14±0.56 | 0.25±0.57 | 0.21±0.98 | 0.44±0.73 | 0.34±0.50 | 0.16±0.79 | 0.29±0.78 | 0.55±0.62 | -0.09 |
| log_prod_of_fro (10) | -0.14±0.58 | 0.56±0.54 | 0.55±0.84 | 0.66±0.59 | 0.68±0.39 | 0.36±0.73 | 0.45±0.73 | 0.84±0.38 | 0.16 |
| euclid_init_norm (11) | 0.23±0.62 | -0.02±0.56 | -0.04±1.00 | 0.39±0.72 | -0.16±0.55 | 0.54±0.60 | 0.53±0.62 | -0.40±0.73 | 0.13 |
| average_ce (12) | -0.50±0.53 | -0.49±0.46 | -0.28±0.96 | -0.76±0.39 | -0.74±0.34 | -0.68±0.49 | -0.74±0.41 | -0.71±0.53 | -0.62 |
| inverse_margin (13) | -0.12±0.70 | -0.28±0.61 | 0.23±0.97 | -0.33±0.77 | -0.28±0.65 | -0.35±0.75 | -0.33±0.77 | -0.45±0.67 | -0.48 |
| prob_margin (14) | 0.50±0.53 | 0.45±0.46 | 0.29±0.96 | 0.76±0.38 | 0.74±0.34 | 0.70±0.48 | 0.75±0.41 | 0.71±0.53 | 0.63 |
| boundary_thickness (15) | -0.38±0.57 | -0.25±0.50 | -0.08±1.00 | -0.59±0.55 | -0.62±0.41 | -0.57±0.58 | -0.65±0.50 | -0.58±0.62 | -0.50 |
| kl_divergence (16) | -0.36±0.60 | -0.03±0.55 | -0.52±0.85 | -0.05±0.80 | -0.39±0.57 | -0.70±0.46 | -0.36±0.73 | -0.53±0.68 | -0.31 |
| local_lip (17) | -0.16±0.66 | 0.15±0.52 | -0.37±0.93 | 0.24±0.76 | -0.06±0.59 | -0.46±0.68 | 0.11±0.79 | 0.11±0.79 | 0.04 |
| pacbayes_flat (18) | 0.32±0.65 | 0.10±0.54 | 0.30±0.95 | 0.10±0.83 | 0.15±0.54 | -0.35±0.71 | -0.31±0.75 | 0.47±0.72 | 0.00 |
| estimated_sharpness (19) | -0.02±0.67 | -0.12±0.55 | 0.26±0.97 | 0.16±0.82 | -0.02±0.61 | -0.41±0.68 | -0.29±0.78 | 0.13±0.81 | -0.26 |
| estimated_inv_sharpness (20) | -0.06±0.66 | -0.14±0.56 | 0.32±0.95 | 0.19±0.82 | 0.02±0.63 | -0.41±0.68 | -0.28±0.77 | 0.03±0.81 | -0.29 |
| average_flat (21) | -0.12±0.56 | 0.12±0.60 | -0.48±0.88 | 0.14±0.79 | -0.09±0.48 | -0.63±0.52 | -0.27±0.76 | -0.22±0.74 | -0.15 |
| x_grad_norm (22) | -0.39±0.63 | -0.22±0.59 | -0.52±0.85 | -0.52±0.68 | -0.65±0.44 | -0.80±0.36 | -0.75±0.47 | -0.70±0.59 | -0.63 |
| w_grad_norm (23) | -0.13±0.65 | -0.14±0.56 | 0.41±0.91 | 0.03±0.84 | -0.14±0.60 | -0.45±0.65 | -0.34±0.74 | -0.03±0.79 | -0.34 |
| average_ce(PGD) (12) | -0.64±0.55 | -0.74±0.41 | -0.70±0.71 | -0.87±0.28 | -0.86±0.26 | -0.80±0.37 | -0.84±0.34 | -0.81±0.45 | -0.76 |
| inverse_margin(PGD) (13) | 0.57±0.48 | -0.13±0.61 | 0.55±0.84 | 0.13±0.82 | 0.39±0.63 | 0.28±0.81 | 0.26±0.81 | 0.10±0.80 | -0.01 |
| prob_margin(PGD) (14) | 0.61±0.58 | 0.57±0.42 | 0.66±0.75 | 0.88±0.27 | 0.85±0.27 | 0.79±0.37 | 0.84±0.34 | 0.79±0.50 | 0.76 |
| pacbayes_flat(PGD) (18) | 0.45±0.62 | 0.29±0.55 | 0.63±0.78 | 0.33±0.78 | 0.45±0.48 | 0.16±0.79 | 0.00±0.83 | 0.65±0.59 | 0.26 |
| estimated_sharpness(PGD) (19) | 0.10±0.66 | 0.03±0.56 | 0.12±0.99 | 0.40±0.71 | 0.17±0.61 | -0.21±0.76 | -0.18±0.79 | 0.29±0.76 | 0.02 |
| estimated_inv_sharpness(PGD) (20) | -0.03±0.66 | -0.01±0.57 | 0.16±0.99 | 0.39±0.73 | 0.17±0.59 | -0.31±0.74 | -0.20±0.79 | 0.20±0.78 | -0.03 |
| x_grad_norm(PGD) (22) | -0.08±0.64 | 0.09±0.61 | -0.48±0.88 | 0.11±0.79 | -0.31±0.55 | -0.61±0.53 | -0.40±0.72 | -0.32±0.75 | -0.19 |
| w_grad_norm(PGD) (23) | -0.09±0.67 | -0.13±0.57 | 0.06±1.00 | 0.25±0.78 | 0.07±0.60 | -0.27±0.76 | -0.17±0.80 | 0.08±0.79 | -0.12 |

Table 11: Numerical results of $\pi_k$ for each measure, along with its corresponding standard deviation. Conditioned on `average_ce(PGD)` $\leq 1.5$. Total $\tau$ is same as in Table 10.

| | Model Architecture | Training Methods | Steps | Optimizer | Batch-size | Aug | Extra-data | Early Stopping | Total $\tau$ |
|---|---|---|---|---|---|---|---|---|---|
| num_params (7) | - | 0.03±0.04 | -0.01±0.09 | 0.03±0.01 | 0.05±0.03 | 0.05±0.07 | 0.04±0.02 | 0.04±0.00 | 0.04 |
| path_norm (8) | 0.64±0.03 | 0.65±0.03 | 0.62±0.02 | 0.64±0.05 | 0.68±0.00 | 0.65±0.03 | 0.63±0.01 | 0.52±0.01 | 0.63 |
| log_prod_of_spec (9) | 0.19±0.25 | -0.09±0.01 | -0.06±0.00 | -0.08±0.02 | -0.07±0.02 | -0.09±0.02 | -0.09±0.00 | -0.08±0.06 | -0.09 |
| log_prod_of_fro (10) | 0.47±0.04 | 0.16±0.05 | 0.17±0.06 | 0.16±0.01 | 0.15±0.04 | 0.19±0.02 | 0.17±0.01 | 0.11±0.01 | 0.16 |
| euclid_init_norm (11) | 0.21±0.07 | 0.15±0.07 | 0.10±0.11 | 0.11±0.01 | 0.16±0.03 | 0.11±0.03 | 0.11±0.02 | 0.17±0.06 | 0.13 |
| average_ce (12) | -0.63±0.04 | -0.70±0.05 | -0.65±0.03 | -0.61±0.05 | -0.62±0.03 | -0.66±0.06 | -0.62±0.05 | -0.56±0.15 | -0.62 |
| inverse_margin (13) | -0.48±0.09 | -0.44±0.22 | -0.47±0.12 | -0.45±0.10 | -0.46±0.12 | -0.47±0.12 | -0.47±0.06 | -0.42±0.28 | -0.48 |
| prob_margin (14) | 0.63±0.03 | 0.71±0.04 | 0.65±0.03 | 0.61±0.04 | 0.62±0.03 | 0.66±0.06 | 0.63±0.05 | 0.57±0.16 | 0.63 |
| boundary_thickness (15) | -0.51±0.03 | -0.61±0.03 | -0.57±0.01 | -0.49±0.07 | -0.50±0.04 | -0.53±0.07 | -0.49±0.09 | -0.45±0.21 | -0.50 |
| kl_divergence (16) | -0.31±0.07 | -0.40±0.14 | -0.13±0.33 | -0.33±0.03 | -0.31±0.11 | -0.25±0.18 | -0.30±0.08 | -0.27±0.13 | -0.31 |
| local_lip (17) | 0.03±0.12 | 0.01±0.23 | 0.17±0.08 | 0.01±0.04 | 0.03±0.08 | 0.13±0.03 | 0.04±0.03 | 0.02±0.03 | 0.04 |
| pacbayes_flat (18) | -0.03±0.19 | -0.01±0.06 | -0.05±0.07 | 0.02±0.01 | -0.00±0.08 | 0.11±0.07 | 0.07±0.06 | -0.16±0.18 | 0.00 |
| estimated_sharpness (19) | -0.26±0.02 | -0.24±0.13 | -0.37±0.01 | -0.25±0.05 | -0.26±0.10 | -0.15±0.16 | -0.22±0.12 | -0.32±0.27 | -0.26 |
| estimated_inv_sharpness (20) | -0.28±0.02 | -0.28±0.14 | -0.42±0.01 | -0.29±0.04 | -0.29±0.06 | -0.18±0.15 | -0.26±0.11 | -0.32±0.27 | -0.29 |
| average_flat (21) | -0.17±0.05 | -0.22±0.13 | 0.02±0.18 | -0.20±0.09 | -0.16±0.05 | -0.04±0.19 | -0.14±0.04 | -0.17±0.01 | -0.15 |
| x_grad_norm (22) | -0.64±0.04 | -0.69±0.13 | -0.55±0.08 | -0.65±0.04 | -0.62±0.08 | -0.58±0.12 | -0.62±0.04 | -0.57±0.14 | -0.63 |
| w_grad_norm (23) | -0.34±0.03 | -0.34±0.11 | -0.48±0.01 | -0.33±0.05 | -0.33±0.08 | -0.26±0.12 | -0.31±0.09 | -0.35±0.27 | -0.34 |
| average_ce(PGD) (12) | -0.77±0.03 | -0.77±0.01 | -0.73±0.03 | -0.76±0.02 | -0.77±0.01 | -0.80±0.02 | -0.77±0.00 | -0.72±0.07 | -0.76 |
| inverse_margin(PGD) (13) | -0.02±0.03 | 0.13±0.11 | -0.14±0.07 | 0.03±0.11 | -0.02±0.05 | -0.01±0.15 | 0.02±0.22 | -0.02±0.31 | -0.01 |
| prob_margin(PGD) (14) | 0.77±0.03 | 0.77±0.01 | 0.73±0.03 | 0.76±0.01 | 0.77±0.02 | 0.80±0.01 | 0.77±0.00 | 0.72±0.08 | 0.76 |
| pacbayes_flat(PGD) (18) | 0.26±0.08 | 0.25±0.07 | 0.25±0.15 | 0.29±0.03 | 0.26±0.07 | 0.34±0.03 | 0.36±0.04 | 0.08±0.20 | 0.26 |
| estimated_sharpness(PGD) (19) | 0.01±0.04 | 0.03±0.09 | 0.06±0.19 | -0.01±0.01 | 0.01±0.13 | 0.14±0.10 | 0.07±0.11 | -0.09±0.17 | 0.02 |
| estimated_inv_sharpness(PGD) (20) | -0.02±0.05 | -0.02±0.08 | 0.00±0.20 | -0.06±0.04 | -0.04±0.08 | 0.10±0.10 | 0.02±0.10 | -0.14±0.20 | -0.03 |
| x_grad_norm(PGD) (22) | -0.20±0.04 | -0.24±0.06 | 0.04±0.28 | -0.25±0.10 | -0.19±0.13 | -0.12±0.18 | -0.18±0.02 | -0.17±0.08 | -0.19 |
| w_grad_norm(PGD) (23) | -0.12±0.04 | -0.08±0.08 | -0.06±0.21 | -0.13±0.01 | -0.11±0.11 | -0.01±0.10 | -0.08±0.08 | -0.19±0.18 | -0.12 |

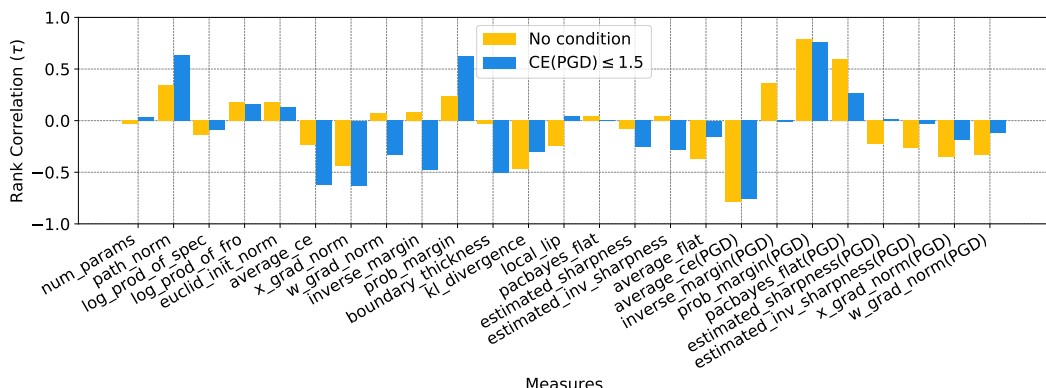

Figure 8: Comparison of total $\tau$ between no condition and `average_ce`(PGD) $\leq 1.5$. Following measures show strong correlation on the condition: `path_norm`, `average_ce`, `x_grad_norm`, `inverse_margin`, `prob_margin`, and `boundary_thickness`.

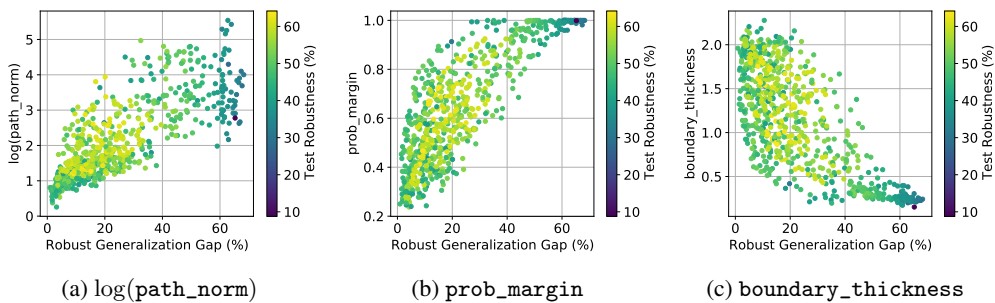

(a) $\log(\texttt{path\_norm})$ \qquad (b) `prob_margin` \qquad (c) `boundary_thickness`

Figure 9: Scatter plot of the measures showing the most increased the total $\tau$ when conditioned on `average_ce`(PGD) $\leq 1.5$.

Table 12: Regression analysis in Fig. 10 for the target variable, the robust generalization gap $g(\boldsymbol{w})$. We summarize their $R^2$ for each measure.

| Measures | $R^2$ | Measures | $R^2$ |
|---|---|---|---|
| num_params (7) | 0.87 | estimated_sharpness (19) | 0.88 |
| path_norm (8) | 0.87 | estimated_inv_sharpness (20) | 0.89 |
| log_prod_of_spec (9) | 0.88 | average_flat (21) | 0.87 |
| log_prod_of_fro (10) | 0.88 | x_grad_norm (22) | **0.91** |
| euclid_init_norm (11) | 0.87 | w_grad_norm (23) | 0.88 |
| average_ce (12) | 0.86 | inverse_margin(PGD) (13) | 0.86 |
| inverse_margin (13) | 0.86 | prob_margin(PGD) (14) | **0.91** |
| prob_margin (14) | 0.87 | pacbayes_flat(PGD) (18) | 0.89 |
| boundary_thickness (15) | 0.86 | estimated_sharpness(PGD) (19) | 0.87 |
| kl_divergence (16) | 0.87 | estimated_inv_sharpness(PGD) (20) | 0.86 |
| local_lip (17) | 0.87 | x_grad_norm(PGD) (22) | 0.88 |
| pacbayes_flat (18) | 0.90 | w_grad_norm(PGD) (23) | 0.87 |

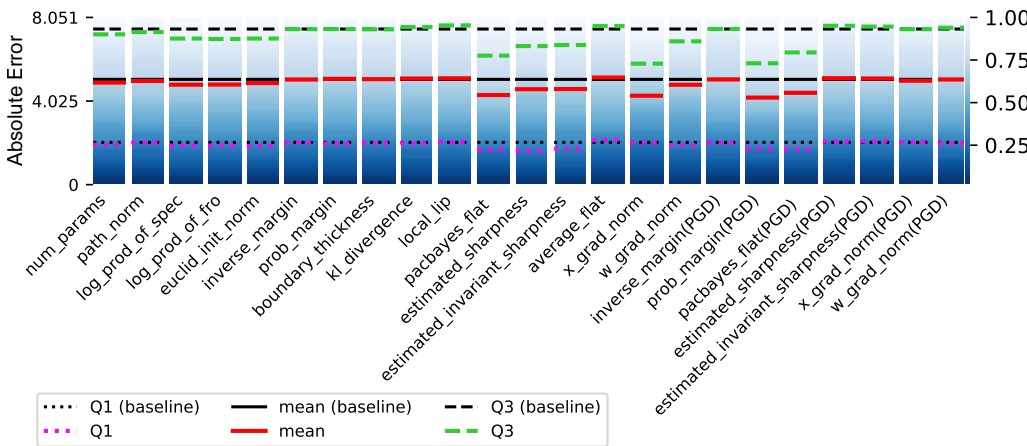

Figure 10: Cumulative distribution of absolute errors for the robust generalization gap estimation is shown. A simple linear regression model is constructed to estimate the robust generalization gap, using `average_ce(PGD)` as the baseline measure, and each measure considered as an additional independent variable. The mean (solid line), Q1 (dotted line), and Q3 (dashed line) are plotted. A lower absolute error indicates that the measure is more effective in estimating the robust generalization gap.

### A.3 Robust Measures with Regression Analysis

In the seminar work of [17], the concept of *an affine oracle* was proposed, which utilizes linear regression to assess the performance of measures. Motivated by this experiment, we conduct a simple linear regression analysis. Specifically, for each measure, we calculate the optimal coefficients and bias to predict the robust generalization gap $g(\boldsymbol{w})$. Considering that `average_ce(PGD)` consistently exhibits the highest correlation across all settings, we use it as a baseline for the regression, i.e., $\beta_1 \times$ `average_ce(PGD)` $+ \beta_0$. We then consider each measure as an independent variable, resulting in a new form of measure $\beta_1 \times$ `average_ce(PGD)` $+ \beta_2 \times$ `measure` $+ \beta_0$. To ensure high predictability, we perform this regression analysis with robust models with `average_ce(PGD)` $\leq 1.5$.

In Table 12, we calculate the coefficient of determination ($R^2$) for the regression analysis to evaluate the extent to which each measure explained the generalization gap. Consistent with the findings in the main paper, `x_grad_norm` and `prob_margin(PGD)` exhibit the highest $R^2$ values.

Additionally, to examine the distributional information of each measure, we plot the cumulative distribution of absolute errors for each model with the mean (solid line) and the interval denoted by the first quartile (Q1, dotted line) and the third quartile (Q3, dashed line). The results are illustrated in Figure 10. We observe that only a few measures, such as `x_grad_norm` and `prob_margin(PGD)`, provided meaningful information about the generalization gap, while the effects of other measures are

Table 13: Total $\tau$ of generated measures from the regression analysis. Both new measures show improved performance in predicting the robust generalization gap.

| Generated Measures | Total $\tau$ |
|---|---|
| $-514.11\times$`x_grad_norm`$-23.87\times$`average_ce(PGD)`$+57.32$ | 0.81 (+0.18) |
| $81.17\times$`prob_margin(PGD)`$+13.52\times$`average_ce(PGD)`$-20.60$ | 0.78 (+0.02) |

Table 14: Total $\tau$ of generated predictors from the regression analysis with forward selection.

| #Measures | Selected Measures | 5-fold $\tau$ (Avg.$\pm$Std.) |
|---|---|---|
| 1 | `average_ce(PGD)` | 0.7229±0.1301 |
| 2 | `x_grad_norm, average_ce(PGD)` | 0.7683±0.1040 |
| 3 | `x_grad_norm, average_ce(PGD), pacbayes_mag_flat(PGD)` | 0.8145±0.0728 |
| 4 | `x_grad_norm, average_ce(PGD), x_grad_norm(PGD), pacbayes_mag_flat(PGD)` | 0.8219±0.0730 |
| All | - | 0.8165±0.0868 |

Table 15: Regression analysis for the target variable, the test robust accuracy $1 - \mathcal{E}(\boldsymbol{w}; \epsilon, \mathcal{D})$. We summarize their $R^2$ for each measure.

| Measures | $R^2$ | Measures | $R^2$ |
|---|---|---|---|
| `num_params` (7) | 0.10 | `estimated_sharpness` (19) | 0.33 |
| `path_norm` (8) | 0.07 | `estimated_inv_sharpness` (20) | 0.41 |
| `log_prod_of_spec` (9) | 0.14 | `average_flat` (21) | 0.12 |
| `log_prod_of_fro` (10) | 0.14 | `x_grad_norm` (22) | 0.09 |
| `euclid_init_norm` (11) | 0.12 | `w_grad_norm` (23) | 0.38 |
| `average_ce` (12) | 0.12 | `inverse_margin(PGD)` (13) | 0.04 |
| `inverse_margin` (13) | 0.05 | `prob_margin(PGD)` (14) | 0.24 |
| `prob_margin` (14) | 0.20 | `pacbayes_flat(PGD)` (18) | 0.48 |
| `boundary_thickness` (15) | 0.20 | `estimated_sharpness(PGD)` (19) | 0.04 |
| `kl_divergence` (16) | 0.13 | `estimated_inv_sharpness(PGD)` (20) | 0.05 |
| `local_lip` (17) | 0.16 | `x_grad_norm(PGD)` (22) | 0.16 |
| `pacbayes_flat` (18) | 0.40 | `w_grad_norm(PGD)` (23) | 0.05 |

negligible. Specifically, `x_grad_norm` and `prob_margin(PGD)` achieve the lowest mean absolute error across all trained models.

In Table 13, we evaluate the effectiveness of predictors generated from the regression analysis. The predictor using `x_grad_norm` achieves an exceptionally strong correlation of 0.81, an increase of 0.18 compared to the previous value in Table 10. The predictor using `prob_margin(PGD)` also exhibits improved performance. These results suggest the potential to effectively predict the robust generalization gap by combining existing measures.

To push further, we conduct a 5-fold evaluation strategy with a linear regression model to predict the robust generalization gap. Specifically, we use the forward selection to identify the most effective set of measures. In Table 14, we present the results of our 5-fold evaluation, reporting the average $\tau$ along with its standard deviation. `average_ce(PGD)` is selected as a prominent predictor, followed by the selection of `x_grad_norm`. Furthermore, our exploration identifies `pacbayes_mag_flat(PGD)` and `x_grad_norm(PGD)` as additional effective measures, resulting in higher average $\tau$ compared to using the entire feature set.

In Table 15, we conduct a linear regression analysis to predict the test robust accuracy, rather than the robust generalization gap. Compared to Table 12, overall values of $R^2$ for each measure are lower. This implies that directly predicting the test robust accuracy might be more difficult than the robust generalization gap.

## A.4  Robust Generalization Gap with AutoAttack [11]

In the main paper, we estimate the robust generalization gap using PGD10. While we acknowledge the potential benefits of stronger attacks, such as AutoAttack [11], we mainly use PGD10 due to the following reasons. Firstly, the computational cost of AutoAttack is substantial. Our experimental design involves training models across diverse adversarial settings and requires adversarial examples for both training and test datasets to estimate the robust generalization gap. AutoAttack takes 10 min per batch for WRN-34-10 on our resources. Since we have 1300 models, we need at least 1 year to obtain all adversarial examples even with 6 GPUs. Secondly, the prevalent usage of PGD among various methods. AT, TRADES, and MART use PGD as a baseline during training and further adopt early-stopping by using PGD on training or validation sets. Lastly, some measures, namely `boundary_thickness` and `local_lip`, rely on PGD adversarial examples for their calculation. As these robustness measures are often computed using PGD, the choice to use PGD for evaluation contributes to consistency across our experiments.

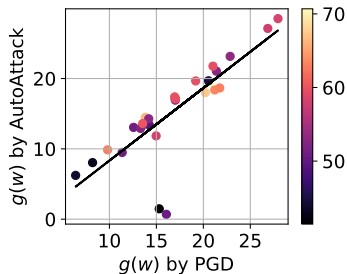

Figure 11: Scatter plot of the robust generalization gap calculated by PGD and AutoAttack. The color of each dot implies the test robust accuracy on CIFAR-10.

However, we here highlight the potential benefits of utilizing AutoAttack in future work. In Fig. 11, we calculate the gap $g(\boldsymbol{w})$ using AutoAttack for 30 models on CIFAR-10 in RobustBench [12]. While the gaps calculated using PGD10 and AutoAttack exhibit an almost linear relationship, there are a few exceptions (2 out of 30): 'Ding2020MMA' [14] and 'Sitawarin2020Improving' [55]. We leave this question open for further exploration.

## A.5  Importance of Model Architecture in Norm-based Measures

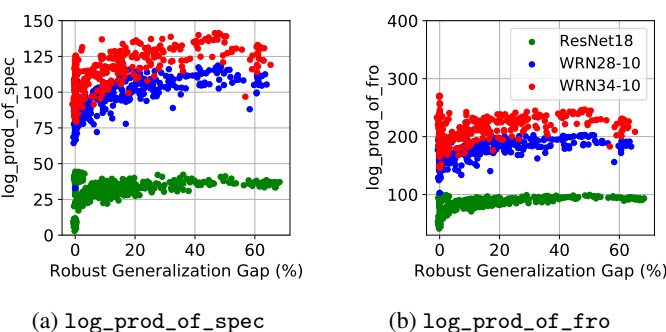

(a) `log_prod_of_spec`          (b) `log_prod_of_fro`

Figure 12: Distributions of norm-based measures for model architectures. For each model architecture, the range of measure extremely varies.

In Fig. 12, we plot `log_prod_of_spec` and `log_prod_of_fro` for each model architecture. We can observe that the range of each measure extremely varies with respect to the used model architecture. When the model architecture is fixed, they show some correlation with the robust generalization gap as described in the main paper.

## A.6  Early Stopping and Estimated Sharpness

In the main paper, we discussed that `estimated_sharpness` exhibits a significant negative correlation when early stopping is not used. Fig. 13 shows the importance of early-stopping for estimated sharpness measures. Compared to Figures 4 and 13b, when early stopping is employed, the correlation approaches zero as shown in Figures 13a and 13c. This results supports the observation of prior study [56] that the importance of early stopping in the analysis of flatness.

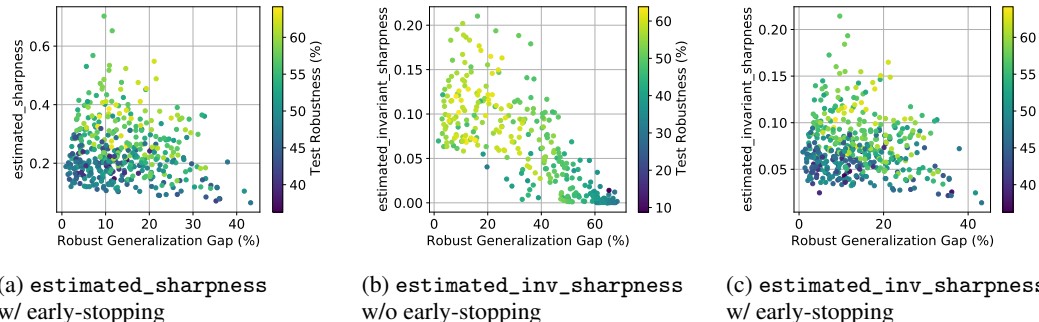

(a) `estimated_sharpness` w/ early-stopping

(b) `estimated_inv_sharpness` w/o early-stopping

(c) `estimated_inv_sharpness` w/ early-stopping

Figure 13: Scatter plot of the estimated sharpness measures for the use of early-stopping. The same condition, `average_ce(PGD)` $< 2$, used as Fig. 4.

## B Measures

In this section, we introduce the concept of each category of measures in the main paper, then explain the details of each measure and their mathematical definitions. Here, we denote $\mathbf{W}_i$ as the weight tensor of $i$-th layer, following [17]. Given the number of layers $d$, the whole trainable parameters are denoted as $\boldsymbol{w} = \text{vec}(\mathbf{W}_1, \mathbf{W}_2, \cdots, \mathbf{W}_d)$.

**Weight-norm.** Based on some theoretical frameworks such as PAC-Bayes [43, 47, 38], weight norm-based measures are expected to be correlated with generalization performance. Among them, the product of Frobenius norm [46] (`log_prod_of_fro`), the product of spectral norm [7] (`log_prod_of_spec`), and path norm (`path_norm`) have been considered as important measures [27, 17]. Furthermore, the distance to the converged weight from the initial weight (`euclid_init_norm`) is also used to estimate the generalization gap [27]. Liu et al. [40] also argues that this distance can be used to judge the difficulty of optimization in adversarial training.
▷ `num_params`.

$$\sum_{i=1}^{d} k_i c_{i-1}(c_i + 1), \tag{7}$$

where $c_i$ is the number of channels and $k_i$ is the kernel size at layer $i$. In the experiments, we calculated `num_params` by adding the number of parameters of all convolutional and linear layers. `num_params` is a fixed value when a model architecture is given.
▷ `path_norm`.

$$\sum_{i} f_{\boldsymbol{w}^2}(\mathbf{1})[i], \tag{8}$$

where $\boldsymbol{w}^2$ is the element-wise square operation and $f(\cdot)[i]$ is the $i$-th logit output of the network. By setting all input variables as 1, this measure captures geometric properties of optimization under scale-invariant characteristics.
▷ `log_prod_of_spec`.

$$\log\left(\prod_{i=1}^{d} \|\mathbf{W}_i\|_2^2\right), \tag{9}$$

where $\|\cdot\|_2$ is a matrix $L_2$-norm, i.e., the largest singular value of each layer.
▷ `log_prod_of_spec`.

$$\log\left(\prod_{i=1}^{d} \|\mathbf{W}_i\|_F^2\right), \tag{10}$$

where $\|\cdot\|_F$ is a Frobenius norm, i.e., the square root of the sum of the squares of the weight matrix.
▷ `euclid_init_norm`.

$$\frac{1}{d}\sum_{i=1}^{d} \|\boldsymbol{w}_i - \boldsymbol{w}_i^0\|_2, \tag{11}$$

where $\boldsymbol{w}_i = \text{vec}(\mathbf{W}_i)$ and the initial weight of $i$-th layer $\boldsymbol{w}_i^0$.

**Margin.** Margins are also actively researched measures to estimate the generalization gap [26]. For instance, the 10-th percentile of the margin values in the output space on the training set (`inverse_margin`) is often used to measure the generalization bound for neural networks [50, 27]. In terms of adversarial robustness, most of attack methods and defense methods utilize the probability margins (`prob_margin`). Yang et al. [67] further proposed a new measure, called boundary thickness (`boundary_thickness`), that is a generalized version of margin and argued that it is highly correlated to the robust generalization gap.

▷ `average_ce`.

$$\mathbb{E}_{\boldsymbol{x},y}\left[\mathcal{L}_{ce}(f(\boldsymbol{x},\boldsymbol{w}),y)\right], \tag{12}$$

where $\mathcal{L}_{ce}$ is the cross-entropy loss.

▷ `inverse_margin`.

$$1/\gamma^2, \tag{13}$$

where $\gamma$ is 10th-percentile of $\{\sigma(f(\boldsymbol{x},\boldsymbol{w}))_y - \max_{i\neq y}\sigma(f(\boldsymbol{x},\boldsymbol{w}))_i\}$ for all $\boldsymbol{x}, y$, with the sigmoid function $\sigma(\cdot)$.

▷ `prob_margin`.

$$\mathbb{E}_{\boldsymbol{x},y}\left[\sigma(f(\boldsymbol{x},\boldsymbol{w}))_y - \max_{i\neq y}\sigma(f(\boldsymbol{x},\boldsymbol{w}))_i\right]. \tag{14}$$

▷ `boundary_thickness`.

$$\mathbb{E}_{\boldsymbol{x}}\left[\|\boldsymbol{x}-\boldsymbol{x}^*\|_2\int_0^1\mathbb{1}\{a<g(\boldsymbol{x},\boldsymbol{x}^*,\lambda)<b\}d\lambda\,\Big|\,\arg\max_i\sigma(f(\boldsymbol{x},\boldsymbol{w}))_i\neq\arg\max_i\sigma(f(\boldsymbol{x}^*,\boldsymbol{w}))_i\right], \tag{15}$$

where $\mathbb{1}\{\cdot\}$ is an indicator function, $g(\boldsymbol{x},\boldsymbol{x}^*,\lambda) = \sigma(f(\lambda\boldsymbol{x}+(1-\lambda)\boldsymbol{x}^*,\boldsymbol{w}))_{\hat{y}} - \sigma(f(\lambda\boldsymbol{x}+(1-\lambda)\boldsymbol{x}^*,\boldsymbol{w}))_{\hat{y}^*}$, $\hat{y} = \arg\max_i\sigma(f(\boldsymbol{x},\boldsymbol{w}))_i$, $\hat{y}^* = \arg\max_i\sigma(f(\boldsymbol{x}^*,\boldsymbol{w}))_i$ and $a, b$ are the hyperparameters that controls the sensitivity of the boundary thickness. Following [67], we find $\boldsymbol{x}^*$ by using PGD10 with $L_2$-norm, $\epsilon=1$, $\alpha=0.2$, then $a=0$, $b=0.75$, and batch size 128. A higher value of `boundary_thickness` implies a larger margin in the output space.

**Smoothness.** Based on prior works [6, 10, 28], a line of work has focused on the smoothness for achieving adversarial robustness in adversarial training. Most simply, the KL divergence between benign and adversarial logits (`kl_divergence`) of TRADES [71] can be regarded as a smoothness regularization due to its logit pairing. Yang et al. [66] investigated the theoretical benefit of local Lipschitzness (`local_lip`) and demonstrated that its value estimated on the test dataset correlates with the robust generalization gap.

▷ `kl_divergence`.

$$\mathbb{E}_{\boldsymbol{x}}\left[\max_{\|\boldsymbol{x}-\boldsymbol{x}*\|\leq\epsilon}\text{KL}(f(\boldsymbol{x},\boldsymbol{w}),f(\boldsymbol{x}^*,\boldsymbol{w}))\right], \tag{16}$$

where KL is KL-divergence and the maximization is conducted by PGD10 with the step-size $2/255$. A lower value of `kl_divergence` implies that a model outputs similar outputs for both benign example and adversarial example.

▷ `local_lip`.

$$\mathbb{E}_{\boldsymbol{x}}\left[\max_{\|\boldsymbol{x}-\boldsymbol{x}*\|\leq\epsilon}\frac{\|f(\boldsymbol{x},\boldsymbol{w})-f(\boldsymbol{x}^*,\boldsymbol{w})\|_1}{\|\boldsymbol{x}-\boldsymbol{x}^*\|_\infty}\right], \tag{17}$$

where the maximization is conducted by PGD10 with the step-size $2/255$. This is the empirical version of the local Lipschitzness, resulting a lower value of `local_lip` implies a smoother model.

**Flatness.** Flatness is a recently focused measure in the generalization domain [43, 30]. Recent works argue that flatter minima yield better generalization performance than sharper minima. The common strategy to achieve flatness is to minimize the estimated sharpness [18, 73], which is the difference between the current loss and the maximum loss for a given neighborhood (`estimated_sharpness`). Kwon et al. [36] investigated the scale-invariant sharpness (`estimated_inv_sharpness`). Note that other diverse concept of estimated sharpness is actively researched in recent works [33, 5, 34]. Adversarial weight perturbation (AWP) [64] also has dramatically improved adversarial robustness by minimizing the loss of perturbed weight. Recently, Stutz et al. [56] has demonstrated that their

proposed measure, average flatness (`average_flat`), is effective for estimating robust generalization gap.

▷ `pacbayes_flat`. Based on PAC-Bayesian framework [43], Jiang et al. [27] proposed a simplified version of PAC-Bayesian bounds as follows:

$$1/\sigma' \tag{18}$$

where $\sigma'$ is the largest value such that $\mathbb{E}_{\boldsymbol{u}}[\mathcal{E}(\boldsymbol{w} + \boldsymbol{u}; \{\boldsymbol{x}, y\}) - \mathcal{E}(\boldsymbol{w}; \{\boldsymbol{x}, y\})] < 0.1$. Here, $u_j \sim \mathcal{N}(0, \sigma')$ and $\mathcal{E}(\boldsymbol{w}; \{\boldsymbol{x}, y\})$ denotes the error on the given set $\{\boldsymbol{x}, y\}$. Thus, a higher value of `pacbayes_flat` implies flatter optimum in terms of error landscape.

▷ `estimated_sharpness`.

$$\mathbb{E}_{\boldsymbol{x}, y} \left[ \max_{\|\boldsymbol{v}\|_2 \leq \rho} \mathcal{L}_{ce}(f(\boldsymbol{x}, \boldsymbol{w} + \boldsymbol{v}), y) - \mathcal{L}_{ce}(f(\boldsymbol{x}, \boldsymbol{w}), y) \right]. \tag{19}$$

Following [73], we calculate the estimated sharpness with a single-step ascent with $\rho = 0.1$. A higher value of `estimated_sharpness` implies a sharper optimum in terms of loss landscape.

▷ `estimated_inv_sharpness`.

$$\mathbb{E}_{\boldsymbol{x}, y} \left[ \max_{\|T_{\boldsymbol{w}}^{-1} \boldsymbol{v}\|_2 \leq \rho} \mathcal{L}_{ce}(f(\boldsymbol{x}, \boldsymbol{w} + \boldsymbol{v}), y) - \mathcal{L}_{ce}(f(\boldsymbol{x}, \boldsymbol{w}), y) \right], \tag{20}$$

where $T_{\boldsymbol{w}}$ is $\|\boldsymbol{w}\|$, i.e., element-wise adaptive sharpness, and $\mathcal{L}_{ce}$ is the cross-entropy loss. Similar to the estimated sharpness, we calculate the estimated invariant sharpness with a single-step ascent with $\rho = 0.1$ [36]. A higher value of `estimated_inv_sharpness` implies a sharper optimum in terms of loss landscape.

▷ `average_flat`.

$$\mathbb{E}_{\boldsymbol{x}, y} \left[ \mathbb{E}_{\boldsymbol{v} \in \mathcal{B}_\rho(\boldsymbol{w})} \left[ \max_{\|\boldsymbol{x} - \boldsymbol{x}^*\| \leq \epsilon} \mathcal{L}_{ce}(f(\boldsymbol{x}^*, \boldsymbol{w} + \boldsymbol{v}), y) \right] - \max_{\|\boldsymbol{x} - \boldsymbol{x}^*\| \leq \epsilon} \mathcal{L}_{ce}(f(\boldsymbol{x}^*, \boldsymbol{w}), y) \right], \tag{21}$$

where $\mathcal{B}_\rho(\boldsymbol{w}) = \{\boldsymbol{w} + \boldsymbol{v} \big| \|\boldsymbol{v}_i\|_2 \leq \rho \|\boldsymbol{w}_i\|_2 \forall \text{ layers } i\}$. We use PGD10 with $\epsilon = 8/255$ for both inner maximizations. Following [56], we take 10 random weight perturbations with $\rho = 0.5$. A higher value of `average_flat` implies a sharper optimum in terms of adversarial loss landscape.

**Gradient-norm.** Gradient-norm with respect to input or weight is also a consistently researched area in terms of generalization. Recently, Zhao et al. [72] also demonstrated that regularizing the gradient norm of weights (`w_grad_norm`) can achieve sufficient improvement on several tasks. Additionally, there are a few works that emphasize the importance of regularizing the gradient norm of weights [53, 44]. The gradient norm of inputs (`x_grad_norm`) also can have underlying correlation between the robust generalization performance. Prior works utilized the input gradient for analyzing adversarially trained models [4] and generating adversarial examples [35, 16].

▷ `x_grad_norm`.

$$\mathbb{E}_{\boldsymbol{x}, y} \left[ \|\nabla_{\boldsymbol{x}} \mathcal{L}_{ce}(f(\boldsymbol{x}, \boldsymbol{w}), y)\|_2 \right]. \tag{22}$$

▷ `w_grad_norm`.

$$\mathbb{E}_{\boldsymbol{x}, y} \left[ \|\nabla_{\boldsymbol{w}} \mathcal{L}_{ce}(f(\boldsymbol{x}, \boldsymbol{w}), y)\|_2 \right]. \tag{23}$$

**Comment on batch normalization fusion** Here, we provide comments on some further discussed things when estimating the above measures. In previous studies [27, 17], it has been observed that considering batch normalization (batch-norm) layers can have an impact on common generalization measures, such as sharpness [15]. To address this issue, the batch-norm layers and other moving statistics were fused with the preceding convolution layers before calculating the values of generalization measures. Thus, when estimating {`num_params`, `path_norm`, `log_prod_of_spec`, `log_prod_of_fro`, `pacbayes_flat`}, we apply batch-norm fusion to all ResNet blocks. However, for certain blocks, such as pre-activation ResNets, where the batch-norm layer is placed at the beginning, the fusion cannot be directly applied. To ensure consistency, we add an identity convolutional layer in front of all batch-norm layers that do not have the preceding convolution layer. While there are various batch-norm fusion (or batch-norm folding) techniques, including those related to

generalization [27, 17, 5], quantization [69, 37, 49], and memory optimization domains [8, 19, 2], there is no precise solution to address this problem in the context of various model structures (ResNets, PreActResNets, ViT, etc.) and activation functions (SiLU, LeakyReLU, etc.), which we leave as a topic for future work.

## C  Training Details

In Section 3, 1,344 models were trained using the CIFAR-10 dataset with $\epsilon = 8/255$. We here provide the detailed training settings. We followed the common settings used in [42, 71, 48, 21].

Given the higher Rademacher complexity [68] and larger sample complexity [54] of adversarial training, data augmentation [21] and the utilization of extra data [9] can significantly improve the adversarial robustness. Therefore, we also considered the impact of augmentation technique, including *RandomCrop* with padding 4 and *RandomHorizontalFlip*, as well as the use of additional data collected by Carmon et al. [9].

Regarding model architectures, we employed three different models: ResNet18 [23], WRN28-10 [70], and WRN34-10 [70]. These models have been widely adopted and serve as benchmarks for evaluating the stability and performance of adversarial training methods. It is worth noting that the majority of models trained on CIFAR-10 in RobustBench [12] consist of WRN28-10 (15) and WRN34-10 (14) among the 63 available models.

For training methods, we considered four different approaches: Standard, AT [42], TRADES [71], and MART [61]. Notably, AT, TRADES, and MART have been shown to outperform other variations by incorporating various training tricks [48] and integrating recent techniques [9, 64]. For all methods, we generated adversarial examples using projected gradient descent (PGD) [42]. During training, a single-step approximation of the inner maximization in Eq. (1) can lead to faster adversarial training, but may suffer from catastrophic overfitting [63, 31]. On the other hand, a large number of steps leads to stable robustness, but requires heavy computational costs. Therefore, we considered both 1 and 10 steps for each adversarial training method.

In terms of optimization, we used SGD with momentum 0.9 and weight decay of $5 \times 10^{-4}$, and a step-wise learning rate decay was performed at epochs 100 and 150 with a decay rate of 0.1. In all the experiments, we trained the models for 200 epochs. As highlighted by Pang et al. [48], the batch size used during adversarial training has been found to affect its performance. Thus, we varied the batch size among {32, 64, 128, 256}. Additionally, we also considered adversarial weight perturbation (AWP) [64], which can improve the robust generalization performance of models. AWP belongs to the class of sharpness-aware minimization methods [18, 36]. This can be formalized as follows:

$$\min_{\boldsymbol{w}} \max_{\|\boldsymbol{x}^{adv}-\boldsymbol{x}\|\leq\epsilon, \boldsymbol{v}\in\mathcal{B}_\rho(\boldsymbol{w})} \mathcal{L}(f(\boldsymbol{x}^{adv}, \boldsymbol{w}+\boldsymbol{v}), y), \tag{24}$$

where $\mathcal{B}_\rho(\boldsymbol{w}) = \{\boldsymbol{w} + \boldsymbol{v} \big| \|\boldsymbol{v}_i\|_2 \leq \rho\|\boldsymbol{w}_i\|_2 \; \forall i\text{-th layer}\}$. As described by Wu et al. [64], $\boldsymbol{x}^{adv}$ is calculated based on the non-perturbed model $f(\boldsymbol{w})$, and a single step of maximization with respect to $\boldsymbol{v}$ is sufficient to improve robustness. We used the best-performing value of $\rho = 5 \times 10^{-3}$ from [64].

