# OpenReview forum: "Fantastic Robustness Measures: The Secrets of Robust Generalization"
_NeurIPS.cc/2023/Conference — NeurIPS 2023 poster_

### Official Review · Reviewer_qzPP · 2023-06-30

**Soundness:** 2 fair
**Presentation:** 2 fair
**Contribution:** 2 fair
**Rating:** 3
**Confidence:** 4

**Summary:**

This paper assesses the correlation between existing measures and the robust generalization gap in various experimental settings. Specifically, the author evaluates the relationship between measures based on weight-norm, margin, smoothness, flatness, gradient-norm, and robust generalization under different conditions such as model architecture, training methods, inner maximization steps, optimizer, batch size, data augmentation, and early stopping. Through extensive experimentation, the author finds that these measures are highly sensitive to training setups and, therefore, not very effective.

**Strengths:**

1. Conduct extensive experiments to explore the correlation between different measures and the robust generalization gap.
2. Present some new findings regarding robust overfitting.


**Weaknesses:**

1. Lack of innovation: The author mainly focuses on existing measures and does not propose new measures.
2. Limited contribution: The author's finding that existing measures are not effective does not provide substantial contributions since the underlying mechanism of robust overfitting remains unclear. If there were a measure capable of accurately evaluating the robust generalization gap, it would also offer a clear direction to the underlying mechanism of robust overfitting.


**Questions:**

Please refer to the comments in Weaknesses.

---

> ### Author Rebuttal · Authors · 2023-08-09
>
> We appreciate the reviewer's insightful assessment of the strengths and weaknesses of our paper. Your feedback provides valuable insights that help us improve the quality and clarity of our work. Below, we provide a detailed response addressing each of the weaknesses  [W#] raised with all references aligned to the main paper:
> ****
>
> **[W1]** We are truly sorry to hear that the innovation of the paper does not match your expectation. Here is a short concrete motivation for studying existing measures. We discover that limitations exist when extending the findings of prior works to practical scenarios due to a restricted set of models [51, 49] and different evaluations [50, 44].
> To this end, we **newly propose a metric** $\pi_k$ in Equation (6) to uncover the effectiveness of measures in the adversarial training framework by training **over 1300 models** and further **verify how and when are measures correlated with the robust generalization gap.**
> To do so, we would very much appreciate it if the reviewer could further comment on what aspects they would judge as the most important to update in the paper.
>
> ****
> **[W2]** The robust generalization gap is *complex* to study directly, particularly within the complexities of real-world datasets [43, 52, 55]. In this paper, we take a significant step towards understanding the robust overfitting by revisiting the measures and uncovering their connections with the robust generalization gap, which was not clearly verified due to the lack of experiments in prior works. We want to emphasize that **other reviewers**, Rev. 2BXc **"This kind of study help evaluate on a more equal footing several of the proposed correlated factors of robust generalisation. I believe this is always helpful for the research community as it helps develop intuition that might lead to novel methods"** and Rev. vGwc **"The paper's findings can inform future experimental and theoretical work on the understanding of the robust overfitting problem,"** agreed with the contribution of our paper.
> Furthermore, our study **provides an original contribution** **by uncovering the high correlation between the 'x_grad_norm' measure and the robust generalization gap**—a novel observation within this research domain. We also have conducted additional experiments in Appendix “A.3 Robust Measures with Regression Analysis”, which introduces the combination of ‘x_grad_norm’ and ‘average_ce(PGD)’ can more effectively predict the robust generalization gap.
>
> ****
> Please refer to general responses for the common questions of other reviewers. **If there are still any unclear points, please feel free to ask detailed questions, we would be happy to provide further explanations.**

---

### Official Review · Reviewer_vGwc · 2023-07-04

**Soundness:** 4 excellent
**Presentation:** 3 good
**Contribution:** 3 good
**Rating:** 7
**Confidence:** 4

**Summary:**

This paper studies the correlation between previously proposed robustness measures and the robustness generalization gap. It studies a large number of models with different architectures and training parameters and shows that some prior beliefs about the usefulness of popular robust generalization measures are not well justified under such comprehensive study.

**Strengths:**

1. The paper comprehensively studies the ability of different robustness measures to predict robust generalization gap and highlights cases where their findings are at odds with prior work and previously held opinions.
2. The methodology and the experiments are clearly and well described.
3. The paper's findings can inform future experimental and theoretical work on the understanding of the robust overfitting problem.

**Weaknesses:**

1. The paper can benefit from further discussion on its findings. For example, the authors say that “margin maximization in adversarial training methods should be revisited” but it isn’t clear why it wouldn’t work. Or that “`boundary_thickness` is more applicable for models trained with AT than to other adversarial training methods” but no intuition as to why that’d be the case is offered. Similarly, they say that “smoothness does not guarantee low robust generalization gap”, which is at odds with the prior work they cite but there is no explanation as to why this discrepancy might occur.
2. Methodologically, the paper seems quite similar to (Jiang et al., 2020). However, (Jiang et al., 2020) seem to have a more comprehensive analysis which attempts at explaining the possible causal factors for the generalization gap. I think this paper can benefit from such a deeper analysis as well.
3. The paragraph starting on line 266 is sends a very conflicting message about the usefulness of `x_grad_norm` for predicting the robust generalization gap. It seems to be saying that it is both highly positively and highly negatively correlated. Perhaps a rewrite for clarity might help.
4. Overall, I found it difficult to understand what are the main takeaways from this work. The conclusion says “Our results suggest practical guidelines for robust generalization” but I am not sure what they are. In several occasions the paper criticizes or is at odds with prior work, but does not provide sufficient evidence as to what are the causes of the discrepancies. I think the paper might benefit from sending a more clear message of what these guidelines are and why some of its findings are at odds with prior work.


References:

Yiding Jiang, Behnam Neyshabur, Hossein Mobahi, Dilip Krishnan, and Samy Bengio. Fantastic generalization measures and where to find them. 2020


**Questions:**

1. On several occasions, it is mentioned that evaluating solely on the test set is undesirable. It’s also used as an explanation for obtaining opposite results to (Yang et al., 2020) but is not fully developed. Why is evaluating the metrics solely on the test set an issue?
2. I found it a bit difficult to understand the purpose of $\pi_k$, the new metric the paper proposes. It seems to be trying to handle cases similar to the Simpson’s paradox, where $k$ fixes a group that has a positive trend when considered in itself, but such that when all groups corresponding to different values of $k$ are considered, the trend becomes reversed. Would this be a valid explanation?
3. (Jiang et al, 2020) show that sharpness-based measures are some of the most predictive ones for the standard generalization gap, while this paper shows that they are not very predictive for the robust generalization gap. How do you reconcile these two seemingly opposing results?
4. In App. A.1, it is argued that there are measures with high correlation with the test robust accuracy but not the robust generalization gap. This is counterintuitive, so what would be possible explanation? High rank correlation with low $R^2$ for linear analysis doesn’t necessarily mean that the measures are not predictive, but rather that the relationship may be highly nonlinear.

**Limitations:**

I've identified no limitations or ethical concerns.

However, here are some suggestions:

1. Maybe instead of writing the mean and confidence intervals in Tabs. 1,2 and 3, you could show horizontal plots of the confidence intervals? That could make it much easier to visually compare the different effects.
2. The measures are only defined in the appendix. I understand that this is for space purposes, but it is difficult to follow your Experiments section without knowing what all these names refer to. I’d recommend saying a couple of words each time you mention a new measure, keeping the formal definitions in the appendix. I think this would improve the readability.
3. Line 199, there’s an unnecessary “to”.
4. Line 699, “maximization” should be “maximizations”

---

> ### Author Rebuttal · Authors · 2023-08-09
>
> We appreciate the reviewer's insightful assessment of the strengths and weaknesses of our paper. Your feedback provides valuable insights that help us improve the quality and clarity of our work. Below, we provide a detailed response addressing each of the weaknesses  [W#], questions [Q#], and limitations [L#] raised with all references aligned to the main paper:
> ****
>
> **[W1]** We sincerely apologize for the lack of discussion on our findings. We certainly agree that more discussions and explanations could benefit the novelty of our paper. To address your concerns, **we have undertaken a comprehensive analysis of 15 additional papers** related to “margin maximization”, “boundary thickness”, and “local Lipschitzness”. **We kindly request the reviewer to refer to our second response to "Weaknesses" in the global response for a more comprehensive explanation.** The forthcoming revised version of our manuscript will include the discussions, which will not only address the concerns but also provide a more robust foundation for our findings. Thank you for your diligence in providing these insights.
>
> ****
> **[W2]** Thank you for your insightful comment regarding the suggestion of experiments on causal factors. In response to your suggestion, **we have performed an additional analysis in Table S1 available in the global response PDF,** focusing on the calculation of the conditional mutual information between robust measures and the robust gap. The cardinality of the set of hyper-parameters $S$ is less than or equal to two (i.e., $|S|\leq2$), in accordance with (Jiang et al., 2020).
>
> We here highlight new findings that align well with the results outlined in the main paper. Specifically, average_ce(PGD) consistently exhibits a significantly high conditional mutual information $\hat{I}(V_\mu, V_g|U_S)$ and shows the highest value of $\kappa(\mu)=0.409$. This indicates the concrete connection between average_ce(PGD) and the robust generalization gap. Similarly, prob_margin(PGD) shows a high value of $\kappa(\mu)=0.387$. local_lip, estimated_sharpness, estimated_inv_sharpness, and average_flat show a low $\hat{I}(V_\mu, V_g|U_S)$ and $\kappa(\mu)$, which is consistent to our observation. Notably, pacbayes_mag_flat(PGD) shows a decent $\kappa(\mu)$, yet it exhibits a relatively low $\hat{I}(V_\mu, V_g|U_S)$ for Steps, which is the important factor in discriminating robust and non-robust models, as discussed in Line 255. x_grad_norm yields a decent $\kappa(\mu)$ compared to other measures, as observed in Table 5 and Figure 5.
>
> We appreciate your feedback, and by incorporating these deeper analyses, we believe our paper now provides a more comprehensive understanding of the underlying causal factors driving the robust generalization gap.
>
> ****
> **[W3]** We apologize for any confusion caused by the phrasing in the paragraph starting on line 266. Allow us to clarify the relationship regarding the usefulness of x_grad_norm in predicting the robust generalization gap. Specifically, x_grad_norm exhibits a high negative correlation with the robust generalization gap compared to other measures. In practical terms, this indicates that models characterized by a lower robust generalization gap tend to have higher x_grad_norm. We will provide clearer paragraphs in the revised version, taking into careful account all the valuable feedback provided by the reviewers.
>
> ****
> **[W4]** In the statement "Our results suggest practical guidelines for robust generalization," the term 'practical guidelines' is intended to emphasize that our study verifies the viability of certain measures with a larger scale of experiments. **As Reviewer 2BXc said that “[…] Many papers claim that Z can be explained by X or Y in some particular setting, while others fail to observe that,”** we believe all our observation (whether certain measures correlated or not correlated to robust generalization) can be helpful for the research community. We have highlighted particularly noteworthy guidelines in blue, which we believe offer substantial insights for future investigations. As this paper primarily aims at an empirical evaluation of measures' effectiveness in predicting the robust generalization gap, our contribution lies more in the realm of empirical findings, similar to [17, 37], rather than being theoretical. **Furthermore, we would like to argue the careful usage of statements such as "Model A is superior to Model B because Model A has a better measure value than Model B," a frequently employed in recent literature.** However, we understand that a lack of discussion on why some of our findings are at odds with prior work. We hope our response to the previous weakness (W1) can resolve this issue. Again, thank you for your valuable suggestion, which will undoubtedly contribute to the depth of our paper.
>
> ****
> **[Q1]** As demonstrated in [23], “the most direct and principled approach for studying generalization in deep learning is to prove a generalization bound which is typically an upper bound on the test error based on some quantity that can be calculated on the training set.” Moreover, it is noteworthy that some researches often use the statement that "superior measure value lead to better generalization" to support the claim that "minimizing (or maximizing) measure value during training results in improved generalization,” which is not always true as demonstrated in [4]. Most importantly, the theoretical analyses in prior work [23, 49, 50] have often used the framework of PAC-learning that basically uses the training set for establishing upper bounds for the generalization gap. In this regard, we believe a comprehensive assessment of measures on the training dataset is essential to avoid wrong conclusions or misuses, such as employing these measures as optimization targets or early-stopping criteria.
>
> ****
> *The remaining questions and limitations will be addressed in the continued official comment due to character limitations.*

---

> > ### Author Response · Authors · 2023-08-10
> > **Rebuttal by Authors [Continue]**
> >
> > ****
> > **[Q2]** Thank you for your interesting interpretation of the new metric $\pi_k$ that we introduced in our paper. We would like to say that your observation is valid, and indeed, $\pi_k$ can be seen as a potential solution for Simpson’s paradox. As noted in Eq. (6), $\pi_k$ has been designed to address cases where specific groups exhibit meaningful trends when considered individually. For instance, in Table 1, the global trend for boundary thickness on Training Methods shows a $\pi$ value of -0.17. However, upon closer look at its high standard deviation, the presence of a distinct trend within a particular group (AT) becomes apparent as shown in Figure 3. We appreciate your insightful perspective and it undoubtedly contributes to enhancing readers' comprehension of the underlying concept of $\pi_k$.
> >
> > ****
> > **[Q3]** First of all, we would like to emphasize the difference between [23] (Jiang & Neyshabur et al., "Fantastic Generalization Measures and Where to Find Them"). In [23], the authors employed customized parameter-efficient neural networks, which are not actively used these days. In contrast, our work employs more practical models such as ResNets, which have been widely used in recent research [11, 46, 44, 50]. We believe this difference in model choice holds significant implications, as recent research [S12, 50] suggested that as neural networks scale up in size, certain phenomena can undergo a reversal. Additionally, recent empirical findings by [4] also demonstrated that sharpness-based measures may not serve as reliable indicators of generalization in modern settings, particularly within the context of the standard training framework. Our observations align with their conclusions that sharpness plays a role as an objective to obtain well-performing models rather than serving solely as a reliable estimator.
> >
> > ****
> > **[Q4]** Indeed, upon delving into prior research [23], which sorely focused on predicting the generalization gap, we were also curious about the relationship between measures and test accuracy itself, rather than the gap. Our motivation for including Appendix "A.1 Estimating Test Robust Accuracy" stemmed from this inquiry, aiming to provide a comprehensive exploration that addresses the interests of readers who share our curiosity.
> >
> > Notably, local Lipschitzness (local_lip) demonstrates a more meaningful total $\tau$ of -0.40 compared to -0.23 in Table 1. It is worth noting that recent research [S15] has highlighted the role of local_lip as an upper bound for the difference in cross-entropy loss between benign and adversarial examples in an adversarial training framework. Consequently, based on this theoretical proof, it would be more accurate to characterize local_lip as correlated with robust accuracy rather than the gap.
> > Most importantly, our focus on investigating the robust generalization gap stems from the consideration that prior research [23, 49, 50] has often emphasized theoretical analyses aimed at establishing upper bounds for the generalization gap, frequently within the framework of PAC-learning. As the field continues to evolve, we anticipate that future work will progressively delve into theoretical analyses offering insights into test accuracies.
> >
> > While we acknowledge the limitations of linear regression in capturing nonlinear relationships between variables, we adopt linear regression since it is the simplest and most intuitive approach for explaining the relationships between measures and the gap. While more complex models may indeed offer greater flexibility in modeling nonlinear interactions, they often come at the cost of interpretability. As we will make our trained models and codes publicly accessible, we expect the research community to explore and implement advanced modeling techniques, offering a promising direction for future research.
> >
> > ****
> > **[L1]** We have attempted to summarize all the experimental results in one table, choosing to report the mean and standard deviation. However, we assure you that we will make every effort to promptly provide any additional experimental results that the reviewers may require. Thus, we reported sample figures in the PDF file in the global response, and we will add all the in Appendix as it requires a large space for all figures.
> >
> > ****
> > **[L2]** We greatly appreciate your feedback regarding the presentation of measures in our paper. Due to the page limitations, we had the challenge posed in providing comprehensive information on each measure within the main paper. To address this concern, we will short descriptions for each measure at the point of their initial mention in Section "4 Experiments." We note that all other typos have also been fixed. Again, thank you for your careful feedback.
> >
> > ****
> > **[L3-L4]**
> > Thank you for bringing the typos to our attention. We have conducted a thorough review of the manuscript and have made the necessary corrections. We again thank you for your detailed review.

---

> > > ### Comment · Reviewer_vGwc · 2023-08-13
> > >
> > > I would like to thank the authors for the additional review, experiments and their comprehensive response. My concerns and questions were mostly addressed, so I increased my score.

---

> > > > ### Author Response · Authors · 2023-08-14
> > > > **Thank you for your feedback**
> > > >
> > > > We are pleased to hear that our response has addressed the majority of your concerns. We truly believe that the quality of the paper has significantly improved through the process of addressing the reviewer's valuable comments. Once again, we would like to express our appreciation for your decision to increase the score, and thank you for your time and expertise!

---

### Official Review · Reviewer_2BXc · 2023-07-06

**Soundness:** 3 good
**Presentation:** 3 good
**Contribution:** 4 excellent
**Rating:** 8
**Confidence:** 4

**Summary:**

The authors propose a large-scale empirical analysis of the impact of several potential indicators of robust generalisation. This paper can be seen as a metaanalysis of many of the indicators proposed in the literature.

**Strengths:**

1. I appreciate a paper aiming to test and/or reproduce at a larger scale of experiments the viability of certain measures as indicators of the robust generalisation gap. Many papers claim that Z can be explained by X or Y in some particular setting, while others fail to observe that. This kind of study helps evaluate on a more equal footing several of the proposed correlated factors of robust generalisation. I believe this is always helpful for the research community as it helps develop intuition that might lead to novel methods.
2. The paper is generally well-written and is quite accessible (see in “Typos and suggestions” questions/recommendations for fixing a part that can be improved; I understood out of intuition what the metric would be but it’d always be better to clarify that part since it is central to your results).


**Weaknesses:**

1. Intro is missing references (in fact, it only has 1 !). References are not optional there in my opinion because they contextualise your paper/motivations. I don’t think refs should come later; plus on your end it’s a matter of adding 10-15 references that are already in your bibliography in section 1, so it can be easily fixed.
2. Only evaluated against PGD on CIFAR10 (that is ok in itself ! It should just be clarified in the claims). Please specify the norm ($L_\infty$) and indicate that (attack choice and dataset) when you summarise your claims (abstract/intro/conclusion) since datasets and the choice of attack likely influences the results.
3. Clarity is generally good but can be improved in 3.3. See “Typos and suggestions”.
4. Correct me if I’m wrong, but it appears the measures are evaluated on the training set, and not a validation set. If so, could that be stated explicitly somewhere ? I believe that’s **very** important (and should also appear clearly in summaries of your results/methodology), especially since many notions/intuitions on flatness may apply better to a validation set, which could lead to different results. See my question, where this could be a possible explanation.


**Typos and suggestions:**
1. L134-135: “each $\Theta_i$ corresponds to a tuple of training parameters” to clarify that they’re not individual parameters. If they are individual parameters, then the sentence “a search space $\Theta$ = …” needs fixing because the right hand side of the equality is not a set (in fact, the right hand side needs fixing anyway to indicate a set).
2. L146: isn’t $\Theta_k$ a tuple of hyperparameters, and $\theta_i$ a particular hyperparameter within such tuples ? I believe eq 5 and/or L146 could benefit from some clarifications. Namely, about what exactly $\theta_k$ vs $\Theta_k$ is.
3. Table 1: I would also indicate in the caption what (PGD) means. Also, indicate what columns correspond to (yes it’s in the text but it’s always better when figures are as self-contained as possible in my opinion).
4. Table 3: average_ce PGD x Model architecture should be bolded too.


**Questions:**

1. I would assume that some measures only make sense within a given model with everything fixed except the weights. For example, flatness vs generalisation. There’s quite some literature in the non-adversarial setting arguing that flatness is correlated with generalisation. However, even within a model, with just the weights varying, it seems not to be the case. Correct me if I’m wrong but this is what Table 4, column “steps” indicates for example (since in that case, two models compared have the same loss landscape but are at a different location in weight space due to not having had as many steps). Do you have intuition why in the robust case this intuition doesn’t hold ? Mine is that it could be tied to some papers arguing that flatness in weight space and input space can be inversely correlated (e.g., “Adversarial Training Makes Weight Loss Landscape Sharper in Logistic Regression” by Yamada et al. 2021). It could also be related to using the training set instead of a validation set to evaluate flatness.

**Limitations:**

I would specify in limitations the choice of dataset, models and attack. And if relevant, the fact all measures were evaluated on the training set if that is correct.

---

> ### Author Rebuttal · Authors · 2023-08-09
>
> We appreciate the reviewer's insightful assessment of the strengths and weaknesses of our paper. Your feedback provides valuable insights that help us improve the quality and clarity of our work. Below, we provide a detailed response addressing each of the weaknesses  [W#], questions [Q#], and limitations [L#] raised with all references aligned to the main paper:
> ****
>
> **[W1]** We appreciate the reviewer's valuable feedback regarding the references in the introduction section. During the refinement of the structure and its flow, the most of sentences containing references are relocated to Related Work. We will ensure the inclusion of references, such as the various phenomena [32, 43, 52, 41], measures [50, 51, 44], and relevant additional sources within Introduction.
>
> ****
> **[W2]** We greatly appreciate your feedback. We agree that the importance of providing clear information regarding the dataset, attack choice, and norm used in our evaluations in the main paper. This will enhance the transparency, readability, and contextual understanding of our paper. We will definitely provide a detailed explanation of these key details in the Introduction and Methodology. We again appreciate the reviewer's thoughtful input.
>
> ****
> **[W3]** Thank you for your detailed remarks. We will update all typos/suggestions and make sure to remove possible other ones.
>
> ****
> **[W4]** We regret any confusion that may have arisen as a result. We would like to clarify that, apart from the weight-norm measure, all other measures are indeed evaluated on the training set, as established in prior works [22, 23, 51]. Although some of the settings are presented in Appendix “B. Measures”, we acknowledge that it has not been mentioned in the main paper. In our revised manuscript, we will take measures to prominently highlight the evaluation of measures on the training set. We will clarify the settings and add detailed settings at the beginning of “3. Experimental Methodology” and “4. Experiments”.
>
> ****
> **[Q1]** First of all, your observation is correct. In the past, the statement that "enhanced flatness corresponds to improved generalisation" had been widely accepted in the deep learning community [13-15]. Indeed, Sharpness-aware minimization have demonstrated superior generalisation performance across various models and tasks, including adversarial training [15, 49]. However, **recent research [4] has challenged this statement** by revealing that flatness does not correlate well with generalisation in standard training framework, particularly within modern practical scenarios. Similarly, our work also finds that flatness does not correlate well with robust generalisation in adversarial training framework. Based on these observations, **we can conclude the following statement – that flatness should be considered as an objective to pursue during optimization, rather than a direct evaluation metric** for predicting a model's generalization performance.
>
> Additionally, we greatly appreciate your insightful observation and the reference you provided, "Adversarial Training Makes Weight Loss Landscape Sharper in Logistic Regression" by Yamada et al. 2021. This paper indeed offers an interesting observation, both theoretically and empirically, on the relationship between perturbations during adversarial training and the sharpness of the loss landscape. Based on their observation, we recognize a potential connection between the observed high values of robust generalization gaps depicted in Figure 1 and the sharpening effect on the loss landscape resulting from larger perturbations during adversarial training. **Indeed, we also confirmed that adversarially trained models tend to exhibit sharper loss landscapes when compared to standard trained models (refer to Figure S2 in the PDF in global response).** We will definitely add a new subsection in our paper.
>
> In assessing the flatness measures on the test set, we conducted an evaluation for both 'estimated_sharpness' and 'estimated_invariant_sharpness’. However, as shown in the below tables, there is **NO** significant correlation between test flatness and the robust generalization gap.
>
> | $\pi_k$ | Model | Method | Steps | Optimizer | Batch | Aug | Semisup | flag | Total $\tau$ |
> | --- | --- | --- | --- | --- | --- | --- | --- | --- | --- |
> | estimated_sharpness(test) | 0.10±0.01 | 0.15±0.03 | 0.12±0.20 | 0.09±0.07 | 0.08±0.04 | 0.14±0.05 | 0.10±0.03 | 0.15±0.03 | 0.09 |
> | estimated_invariant_sharpness(test) | 0.12±0.01 | 0.16±0.05 | 0.09±0.18 | 0.13±0.06 | 0.12±0.04 | 0.17±0.02 | 0.13±0.00 | 0.15±0.05 | 0.12 |
>
> Certainly, our analyses could be extended to test examples (or validation examples), but for our study, we chose to focus on train examples. This choice aligns with the approach outlined in [23], which argues that “the most direct and principled approach for studying generalization in deep learning is to prove a generalization bound which is typically an upper bound on the test error based on some quantity that can be calculated on the training set.” Moreover, it is noteworthy that some researches often use the statement that "superior measure value lead to better generalization" to support the claim that "minimizing (or maximizing) measure value during training results in improved generalization,” which is not always true as demonstrated in [4]. Most importantly, the theoretical analyses in prior work [23, 49, 50] have often used the framework of PAC-learning that basically uses the training set for establishing upper bounds for the generalization gap. In this regard, we believe a comprehensive assessment of measures on the training dataset is relatively important to avoid wrong conclusions or misuses, such as employing these measures as optimization targets or early-stopping criteria.
>
> ****
> **[L1]** We sincerely hope that the responses provided above address the outlined weaknesses and limitations. Thank you once again for your insightful assessment and guidance.

---

> > ### Comment · Reviewer_2BXc · 2023-08-14
> >
> > I would like to thank the authors for their detailed reply.
> >
> > > [W2] We greatly appreciate your feedback. We agree that the importance of providing clear information regarding the dataset, attack choice, and norm used in our evaluations in the main paper. This will enhance the transparency, readability, and contextual understanding of our paper. We will definitely provide a detailed explanation of these key details in the Introduction and Methodology. We again appreciate the reviewer's thoughtful input.
> >
> > I still believe this should be in the abstract too, given that several works show how loosely correlated robustness against different norms or types of attacks can be (it could even be a stated limitation).
> >
> > >  Indeed, we also confirmed that adversarially trained models tend to exhibit sharper loss landscapes when compared to standard trained models (refer to Figure S2 in the PDF in global response). We will definitely add a new subsection in our paper.
> >
> > That will be an interesting addition indeed.
> >
> > > Certainly, our analyses could be extended to test examples (or validation examples), but for our study, we chose to focus on train examples. This choice aligns with the approach outlined in [23], which argues that “the most direct and principled approach for studying generalization in deep learning is to prove a generalization bound which is typically an upper bound on the test error based on some quantity that can be calculated on the training set.” Moreover, it is noteworthy that some researches often use the statement that "superior measure value lead to better generalization" to support the claim that "minimizing (or maximizing) measure value during training results in improved generalization,” which is not always true as demonstrated in [4]. Most importantly, the theoretical analyses in prior work [23, 49, 50] have often used the framework of PAC-learning that basically uses the training set for establishing upper bounds for the generalization gap. In this regard, we believe a comprehensive assessment of measures on the training dataset is relatively important to avoid wrong conclusions or misuses, such as employing these measures as optimization targets or early-stopping criteria.
> >
> > At the risk of saying something obvious, generally, the test set should be used for general final conclusions about metrics, the validation set for hyperparameter selection and early stopping (yes it's part of training but the weights aren't trained on it, it's a proxy for a test set), and of course the training set for direct optimisation / parameter tuning. The reason why is that the underlying assumption behind many of the theoretical claims mentioned is that everything is i.i.d. or equivalently that the training data is representative of the test data, and a second point is that this assumes the performance of a model doesn't come (partly) from memorisation. Nowadays, with overparametrising becoming the norm, the $\simeq 0$ training loss is evidently poorly indicative of the test loss.

---

> > > ### Author Response · Authors · 2023-08-16
> > >
> > > Thank you for your response to our rebuttal.
> > >
> > > > I still believe this should be in the abstract too, given that several works show how loosely correlated robustness against different norms or types of attacks can be (it could even be a stated limitation).
> > >
> > > We will certainly include the information about the norm ($L_\infty$) and other relevant settings in the abstract and limitation.
> > >
> > > > At the risk of saying something obvious, generally, the test set should be used for general final conclusions about metrics, the validation set for hyperparameter selection and early stopping (yes it's part of training but the weights aren't trained on it, it's a proxy for a test set), and of course the training set for direct optimisation / parameter tuning. The reason why is that the underlying assumption behind many of the theoretical claims mentioned is that everything is i.i.d. or equivalently that the training data is representative of the test data, and a second point is that this assumes the performance of a model doesn't come (partly) from memorisation. Nowadays, with overparametrising becoming the norm, the 0  training loss is evidently poorly indicative of the test loss.
> > >
> > > We sincerely appreciate your comment, and it has prompted us to rethink our methodology. In particular, your comment *"The reason why is that the underlying assumption behind many of the theoretical claims mentioned is that everything is i.i.d. or equivalently that the training data is representative of the test data,"* highlights a crucial aspect that we had overlooked. We now acknowledge the potential importance of analyzing the test data with measures. In light of your feedback, we have revisited our code and have initiated the process of conducting the same analysis on the test set. We anticipate having the results available before the rebuttal stage, but please note that it may require some time. We again appreciate your additional guidance.

---

> > > > ### Comment · Reviewer_2BXc · 2023-08-17
> > > >
> > > > I appreciate the significant efforts made by the authors, and will increase my score (7->8).

---

> > > > > ### Author Response · Authors · 2023-08-17
> > > > >
> > > > > We would like to extend our gratitude for your decision to revise the score upward, and we sincerely thank you for the time and expertise you have dedicated to this review process!

---

### Official Review · Reviewer_PVgT · 2023-07-10

**Soundness:** 2 fair
**Presentation:** 2 fair
**Contribution:** 2 fair
**Rating:** 6
**Confidence:** 4

**Summary:**

The paper conducts a large scale correlation analysis for adversarially trained models to identify which existing robustness-related measures are predictive of robust overfitting. The correlation analysis is performed over large set of relevant dimensions (including model architectures, training procedures, attack strengths, optimizers, batch sizes, etc.). The overall experimental methodology seems fine.

**Strengths:**

The experiments uncover some interesting findings regarding which metrics can be predictive of robust generalization.

A definite strength of this work is that all code and trained models are made available online -- this will enable the adversarial ML community at large to build on the analysis explored herein with new measures, training methods/settings/etc.

**Weaknesses:**

1. A major significant weakness of this work is its title. The title is non-descriptive and incendiary. For an academic publication (i.e. not a blog post, news article or social media post) this seems wholly unacceptable. Please update it to something suitably descriptive instead.

2. How related this work is with reference [23] (Jiang & Neyshabur et al., "Fantastic Generalization Measures and Where to Find Them") is severely downplayed. Essentially, this work follows almost the exact same playbook as [23]. A subsection on the relationship of this work with [23] should be included in this paper. For example, a small step in this direction is section 3.3 which already points towards the evaluation metrics being wholly adopted from [23] while changing the notation.

3. The attack used for measuring the robustness of trained models on the test set is just PGD-10 and this is not strong enough for a proper evaluation of robustness. AutoAttack (at least), and e.g., Square and Multi-Targeted attacks should've been used instead to better approximate the models' adversarial robustness. To root out training procedures that are affected by gradient obfuscation, using just a simple gradient-based attack (with just 10 steps; no restarts; etc.), like PGD-10 for evaluation, is insufficient. What slightly ameliorates this point is the inclusion of analysis of models from RobustBench (e.g. Figure 5).

4. Only when models are actually robust on the train set it is meaningful to measure the robustness gap; including the bottom left part of Figure 1 seems like it would just introduce noise into the whole estimation process.

5. Two important sections, including the Key Findings and Section 3.3 (which introduces the evaluation metrics) should be re-written much more clearly. For example, key finding 1 states "traditional metrics may be ineffective due to the high sensitivity with respect to training setups" -- this is very unclear? What are traditional metrics? Saying that something "may be ineffective" is so imprecise that it becomes not useful for the reader. In key finding 3, it is said that "flatness-based measures [...] perform poorly and even sharper minima can exhibit a lower robust generalisation gap" -- what does it mean for sharper minima to exhibit a lower robust generalisation gap? Do you mean something like: "neural networks which have sharper minima exhibit a lower generalisation gap in 70% of cases we evaluated?". Please rephrase. For section 3.3, I had to carefully read [23] to try to be able to parse what section 3.3 is stating -- please clarify all the writing from 147-159; e.g., the point in lines 144-145 is completely unclear by itself; the point in lines 158-159 has typos and is unclear; similarly, what does "generalization gap varies sensitively with respect to training setups" mean? Could you motivate why the pi_k metric is effective (line 152)? An explanation of why this might be the case only shows up in section 4.

**Questions:**

Do the key findings hold when considering only trained models which have high train & test robustness?

Table 1 is not very easy to read at a high-level; have you attempted colour-mapping it (i.e. like a heatmap)?

Have you tried training a classifier (e.g. just an MLP) using all measures as input features to predict a discretized robust generalization gap? Depending on how well the classifier generalizes, this could be useful for as a direct optimization proxy to improving the predicted robust generalization gap.

It seems a bit awkward to not qualify "measures" with a prefix; for example, in [32] the considered measures are named "complexity measures". E.g., have you considered something like "robustness measures", "proxy measures" etc.?

See also the section on weaknesses above.

**Limitations:**

Limitations are discussed briefly.

---

> ### Author Rebuttal · Authors · 2023-08-09
>
> We appreciate the reviewer's insightful assessment of the strengths and weaknesses of our paper. Your feedback provides valuable insights that help us improve the quality and clarity of our work. Below, we provide a detailed response addressing each of the weaknesses  [W#] and questions [Q#] raised with all references aligned to the main paper:
> ****
>
> **[W1]** We are sorry to hear that the title of the paper is unacceptable. The initial title was motivated by [23] (Jiang & Neyshabur et al., "Fantastic Generalization Measures and Where to Find Them"), but we understand the need for a more accurately descriptive title that better reflects the content and objectives of our research. **We have taken this feedback seriously and are in the process of revising the title** to ensure that it appropriately represents the scholarly nature of our work. One of the potential titles that we are currently considering is 'Evaluating Generalization Measures in Adversarial Training Framework.' We believe this title encapsulates the core focus of our research, highlighting the evaluation of generalization measures within the context of an adversarial training framework. However, if you have any further suggestions or ideas for a more fitting title, we would greatly appreciate your input.
>
> ****
> **[W2]** We appreciate the reviewer's feedback regarding the relationship between our work and the reference [23]. Indeed, our study draws inspiration from the valuable insights presented in [23], which conducted an evaluation of complexity measures in the standard training framework. However, due to space limitations in the main paper, we regrettably could not thoroughly delve into the connection between our work and [23]. In response to the reviewer's suggestion, **we introduce the following new subsection that explicitly addresses the connection between our study and [23] based on our answer to the first weakness in the global response:**
>
>
> \subsection{Comparison to Jiang et al. [23]}
> The pioneering study [23] explored the empirical correlations between complexity measures and generalization, with a primary focus on the standard training framework. Our main contribution is delving specifically into the realm of robustness measures within the adversarial training framework—a context having different generalization tendencies even with the same measures and experimental settings, as demonstrated by prior research [43, 52]. Of significance is the observation that the metric $\psi_k$ proposed in [23] has limitations in accurately capturing the effectiveness of robustness measures due to its high variance with respect to training setups. By introducing a new metric, our work enhances the understanding of when and how robustness measures correlate with robust generalization. Moreover, while Jiang et al. [23] employed customized parameter-efficient neural networks, we adopt widely-used model architectures such as ResNets, thereby providing insights that are not only relevant to recent research but also offer more practical implications.
>
> ****
> **[W3]** While we acknowledge the potential benefits of AutoAttack, the constraints outlined above guided our decision to use PGD-10. **We believe that Appendix “A.4 Robust Generalization Gap with AutoAttack”**, which demonstrates the comparison between the robust generalization gaps calculated using PGD and AutoAttack, **can help alleviate some of the concerns you've raised.** However, to address your concern in detail, we here explain three reasons why we primarily used PGD-10.
>
> **Firstly, the computational cost of AutoAttack is substantial.** Our experimental design involved training models across diverse adversarial settings and required adversarial examples for both training and test datasets to estimate the robust generalization gap. However, with AutoAttack, it takes 10 min/batch for WRN-34-10 on our resources. **Since we used 1300 models, we need at least 1 year to obtain all adversarial examples even with 6 GPUs.** Thus, the resource-intensive nature of AutoAttack led to impractical computation times, making it infeasible to execute extensive experiments.
>
> **Secondly, the prevalent usage of PGD as a baseline during training** among various methods. Indeed, all popular adversarial training methods, such as AT, TRADES, and MART, use PGD as a baseline during training, and thus early-stopping (or other similar techniques) is adopted with PGD on training or validation sets. This led us to consider PGD as a more practical choice for providing practical insights.
>
> **Lastly, the specific robustness measures we employed, namely boundary_thickness and local_lip, rely on PGD adversarial examples for their calculation.** As these robustness measures are often computed using PGD, the choice to use PGD for evaluation contributes to consistency across our experiments.
>
> ****
> **[W4]** In the main paper, we mainly conducted experiments on all possible models to comprehensively validate the robustness measures across a wide range of models. However, we also have shared a curiosity about the implications of our analysis on models with high robustness. **To verify this, we have provided “A.2. Focusing on Adversarially Robust models” in Appendix.** Specifically, we selected models by conditioning on ‘average_ce(PGD)’ ≤ 1.5. Upon analyzing these adversarially robust models, we observed distinctive behaviors in certain robustness measures (’path_norm’, ‘average_ce’, ‘x_grad_norm’, ‘inverse_margin’, ‘prob_margin’, and ‘boundary_thickness’). For example, margin-based measures displayed more pronounced negative correlations, while the effectiveness of ‘boundary_thickness’ in estimating the robust generalization gap became effective. Please refer to Appendix A.2. for more detailed explanations and graphical representations of our findings.
>
> ****
> *The remaining weakness and questions will be addressed in the continued official comment due to character limitations.*

---

> > ### Author Response · Authors · 2023-08-10
> > **Rebuttal by Authors [Continue]**
> >
> > ****
> > **[W5]** We apologize for the unclear points and thank you very much for reading carefully. **To address your concerns, we thoroughly revised the sections and sentences you highlighted.** For instance, we have revised the following sentences as follows:
> >
> > * Original: "Evaluating measures under the adversarial training framework with traditional metrics may be ineffective due to the high sensitivity with respect to training setups."
> > Revised: "Given the high sensitivity of the robust generalization gap with respect to training setups, calculating the expectation of the rank correlation across a wide range of training setups yields a high variance and fails to capture a trend that appears in several groups of data."
> > * Original: "Flatness-based measures, such as estimated sharpness, generally perform poorly and even sharper minima can exhibit a lower robust generalization gap."
> > Revised: "Flatness-based measures, such as estimated sharpness, generally perform poorly in predicting the robust generalization gap. Rather, they show a negative correlation, indicating that models with sharper minima tend to exhibit a lower robust generalization gap."
> >
> > **Finally, we would like to clarify the motivation and effectiveness of $\pi_k$.** The notion of "generalization gap varies sensitively with respect to training setups" refers to the phenomenon where the range of the robust generalization gap can significantly differ based on the specific configuration of training parameters. This characteristic, distinct from the standard training framework [23], is evident in Figure 1, where the robust generalization gap exhibits a wide range of values, spanning from 0 to 65, across various training setups. This aligns with prior findings [37] that the performance of adversarial training is heavily affected by diverse hyperparameters, such as optimizers, batch size, and early stopping.
> >
> > Due to this distinct characteristic of adversarial training,  the metric $\psi_k$ proposed by [23] exhibits extremely high variance as shown in Table 1. This high variance of $\psi_k$ raises concerns about the reliability of its connections and the potential for misleading conclusions. As Rev. vGwc mentioned, this phenomenon can be explained by Simpson's paradox, where $\psi_k$ might provide misleading results or struggle to capture meaningful correlations between robustness measures and the robust generalization gap.
> >
> > To address these concerns, we introduced the metric $\pi_k$ to offer a deeper understanding of correlations between robustness measures and the robust generalization gap. Notably, we have observed that $\pi_k$ demonstrates significantly lower variance compared to $\psi_k$ (Table 2), and it enhances our ability to identify conditions that yield high rank correlations for specific measures (Figure 3).
> >
> > We will update all the responses in the revised version. We sincerely thank you for helping us improve the overall quality of our work.
> >
> > ****
> > **[Q1]** Please refer to the response to [W4].
> >
> > ****
> > **[Q2]** We appreciate the reviewer's suggestion to enhance the readability of Table 1. While we have tried to visualize well Table 1, the extensive information contained in the table poses a challenge for effective color representation. We believe that, given the comprehensive nature of the data we are presenting, visualizing it might introduce complexity. One approach we are considering is the incorporation of bar plots, which could provide a clearer representation of the data distribution and relationships. Thus, we reported some figures in the accompanying PDF in this global response, and we will add all the in Appendix as it needs large space for figures.

---

> > > ### Author Response · Authors · 2023-08-10
> > > **Rebuttal by Authors [Continue]**
> > >
> > > ****
> > > **[Q3]** In Appendix 'A.3 Robust Measures with Regression Analysis', we have conducted a linear regression analysis to explore the predictive potential of the combinations of robustness measures on the robust generalization gap.
> > > Given that 'average_ce(PGD)' consistently demonstrated the highest correlation with the robust generalization gap across diverse settings, we incorporated each measure as an independent variable.
> > > Notably, our experimentation revealed that a combination of 'x_grad_norm' and 'average_ce(PGD)' yielded the most compelling performance in predicting the robust generalization gap.
> > >
> > > To push further, we extend our exploration with an additional experiment, wherein we employed a 5-fold evaluation strategy with a linear regression model to predict the robust generalization gap. During the experiment, We also consider feature selection. Specifically, we employed forward selection to identify the most effective set of measures. In the subsequent table, we present the results of our 5-fold evaluation, reporting the average $\tau$ along with its standard deviation.
> > >
> > > | #Measures | Selected Measures | 5-fold $\tau$ (Avg.±Std.) | Increment |
> > > | --- | --- | --- | --- |
> > > | 1 | ['average_ce(PGD)'] | 0.7229±0.1301 |  |
> > > | 2 | ['x_grad_norm', 'average_ce(PGD)'] | 0.7683±0.1040 | +0.0454 |
> > > | 3 | ['x_grad_norm', 'average_ce(PGD)', 'pacbayes_mag_flat(PGD)'] | 0.8145±0.0728 | +0.0462 |
> > > | 4 | ['x_grad_norm', 'average_ce(PGD)', 'x_grad_norm(PGD)', 'pacbayes_mag_flat(PGD)'] | 0.8219±0.0730 | +0.0073 |
> > > | All | - | 0.8165±0.0868 |  |
> > >
> > > As we observed, 'average_ce(PGD)' is selected as a prominent predictor,  followed by the selection of 'x_grad_norm'. This result is consistent with Tables 12 and 13. Furthermore, our exploration identifies 'pacbayes_mag_flat(PGD)' and 'x_grad_norm(PGD)' as additional effective measures, resulting in higher average $\tau$ compared to using the entire feature set.
> > >
> > > We sincerely appreciate your constructive input, which motivated us to conduct these additional analyses. We will surely include this discussion in a revised version of the paper.
> > >
> > > ****
> > > **[Q4]** We appreciate your suggestion regarding the terminology used for the measures. Your suggestion of using the term 'robustness measures' is well taken, as it provides a clearer context for the nature and intent of the metrics under evaluation. In our revised version, we will adopt this terminology consistently throughout the manuscript to ensure greater clarity and precision. Thank you.

---

### Author Rebuttal · Authors · 2023-08-09

**Dear all,**

We would like to thank the editor and the reviewers for their careful comments and suggestions.

We summarize the reviews according to our own perspective.
****
**Strengths.**

We are glad that Reviewers PVgT, 2BXc, and vGwc found that our results **“uncover some interesting findings regarding which metrics can be predictive of robust generalization”** and **“is always helpful for the research community as it helps develop intuition that might lead to novel methods.”** Additionally, Reviewers 2BXc and vGwc also highlighted that **“the paper is generally well-written”** and is rooted in **“clearly and well-described”** methodology and experiments.

$   $

**Weaknesses.**

We have welcomed all reviews and did our best to carefully addressed every concern.

Specifically, the reviewers raised the following common concerns.

1. Lack of detailed explanations on the relationship to [23] and distinctions between our study and the findings of [23]. (Rev. PVgT and vGwc)
2. More discussion of results and contributions. (Rev. vGwc and qzPP)
3. Visibility of Tables, which are not very easy to read at a high-level. (Rev. PVgT and vGwc)

**Our answer to the common concerns raised by the reviewers:**

1. We carefully revisited [23](Jiang & Neyshabur et al., "Fantastic Generalization Measures and Where to Find Them") and we here introduce a summarization of the differences between our study and the findings of [23].
To summarize, **(i) our work delves into the adversarial training framework**, diverging from the standard training framework in [23]. Furthermore, **(ii) our choice of widely-used models such as ResNets**, in contrast to the customized parameter-efficient structures of [23]. Importantly, **(iii) we introduce the novel metric $\pi_k$**, addressing the limitations of the metric $\psi_k$ from [23], facilitating a precise assessment of the correlations between measures and robust generalization.
We will add a new subsection that explicitly addresses clarification of the relationship to [23] in the revised version.
2. To augment the comprehensiveness of our results and their contributions, **we conducted an in-depth review of additional 15 papers (referenced as [S#])** ━will be updated in the continued comment due to character limitations━ related to “margin maximization”, “boundary thickness”, and “local Lipschitzness”.
   * Margin maximization: According to prior works [S1, S2], models trained with CrossEntropy serve as max-margin predictors by optimizing a lower bound of the margin within both standard and adversarial training frameworks. However, recent studies have highlighted that maximizing the margin might not necessarily be the optimal objective in adversarial training due to intricate gradient flow dynamics [S3] and the non-cognitive concept of using predicted probabilities [S4]. Moreover, in recent work [S5, S6], despite the similar robustness of TRADES and AT, their margin distributions on benign and adversarial examples are extremely different. This implies that the margin cannot be the sole determinant of adversarial robustness. Our findings also support these observations by revealing that margin alone does not correlate well with robust generalization; in fact, its excessive pursuit of margin maximization can potentially have an adverse impact on robust generalization. Considering other recent studies [S7, S8], the margin maximization should be accompanied by a consideration of other factors such as weight regularization or gradient information.
   * Boundary thickness: Boundary thickness often serves as a supportive measure to validate the robustness of new training algorithms [S10]. However, empirical validation of its efficacy remains sparse, primarily due to the limited focus on AT in the original paper. Intriguingly, our comprehensive experiment reveals that boundary thickness exhibits a relatively low correlation with the robust generalization gap within models trained by TRADES or MART. We believe that this is caused by the limitation of the margin because boundary thickness also sorely depends on the probability margin. Considering that AT shows a more prominent margin distribution compared to TRADES [S5], this margin-based measure, boundary thickness might be more aligned with AT —where robustness is attained through margin maximization—than with other training methodologies. Indeed, as shown in Figure 3, AT is placed as an outlier for both prob_margin and boundary_thickness. Consequently, our study cautions against the assumption that a higher boundary thickness implies better robust generalization. We emphasize that the concurrent work [S11] also points out that boundary complexities such as boundary thickness are highly abstract.
   * local Lipschitzness: While the foundational work [S12] demonstrates that imposing local Lipschitzness leads to better generalization in linear classification, recent research [50] presents an opposing perspective, suggesting that within neural networks, local Lipschitzness might hurt robust generalization. However, it's noteworthy that the conclusions drawn in [50] are built on limited models of fewer than 20 and are sorely based on test examples, warranting further investigation for comprehensive validation. In our large experiment, we cannot observe that local Lipschitzness itself negatively affects robust generalization. Moreover, in Appendix Tables 8 and 9, we observe that local Lipschitzness is also not perfectly aligned with robust accuracy in contrast to [50, S15]. These findings are consistent with recent works [S13, S14], which highlight the importance of model architecture or weight norms when evaluating models with local Lipshitzness.
3. To provide easy visualization of Tables, we will add the visualization of our tables by providing multiple point plots with error bars for each table in the Appendix. **Sample plots are available in the accompanying PDF in this “global” response.**

**↓Accompanying PDF**

---

> ### Author Response · Authors · 2023-08-10
> **Additional papers [S#]**
>
> [S1] Mitros, John, and Brian Mac Namee. "On the Importance of Regularisation and Auxiliary Information in OOD Detection." International Conference on Neural Information Processing. Cham: Springer International Publishing, 2021.
>
> [S2] Ding, Gavin Weiguang, et al. "MMA Training: Direct Input Space Margin Maximization through Adversarial Training." *International Conference on Learning Representations*. 2019.
>
> [S3] Vardi, Gal, Gilad Yehudai, and Ohad Shamir. "Gradient methods provably converge to non-robust networks." *Advances in Neural Information Processing Systems* 35 (2022): 20921-20932.
>
> [S4] Adachi, Hiroki, et al. "Revisiting Instance-Reweighted Adversarial Training." (2022).
>
> [S5] Kim, Hoki, et al. "Bridged adversarial training." *arXiv preprint arXiv:2108.11135* (2021).
>
> [S6] Wang, Haotao, et al. "Removing batch normalization boosts adversarial training." *International Conference on Machine Learning*. PMLR, 2022.
>
> [S7] Liu, Ziquan, and Antoni B. Chan. "Boosting adversarial robustness from the perspective of effective margin regularization." *arXiv preprint arXiv:2210.05118* (2022)
>
> [S8] He, Zhengbao, et al. "Investigating Catastrophic Overfitting in Fast Adversarial Training: A Self-fitting Perspective." *Proceedings of the IEEE/CVF Conference on Computer Vision and Pattern Recognition*. 2023.
>
> [S10] Li, Yaxin, et al. "Enhancing Adversarial Training with Feature Separability." *arXiv preprint arXiv:2205.00637* (2022).
>
> [S11] Piwek, Paweł, Adam Klukowski, and Tianyang Hu. "Exact Count of Boundary Pieces of ReLU Classifiers: Towards the Proper Complexity Measure for Classification." *arXiv preprint arXiv:2306.08805* (2023).
>
> [S12] Xu, Huan, Constantine Caramanis, and Shie Mannor. "Robustness and Regularization of Support Vector Machines." *Journal of machine learning research* 10.7 (2009).
>
> [S13] Nern, Laura Fee, and Yash Sharma. "How Adversarial Robustness Transfers from Pre-training to Downstream Tasks." *arXiv preprint arXiv:2208.03835* (2022).
>
> [S14] Liang, Youwei, and Dong Huang. "Large norms of CNN layers do not hurt adversarial robustness." *Proceedings of the AAAI Conference on Artificial Intelligence*. Vol. 35. No. 10. 2021.
>
> [S15] Huang, Hanxun, et al. "Exploring architectural ingredients of adversarially robust deep neural networks." *Advances in Neural Information Processing Systems* 34 (2021): 5545-5559.

---

### Decision · Program_Chairs · 2023-09-21

**Decision:**

Accept (poster)

**Comment:**

The paper critically investigates the correlation between the robust generalization gap and multiple “robustness measures” studied in prior literature. It provides a medium-to-large scale study on 1,300 trained models, and it outlines robustness measures from multiple methodologies, including gradient-based, margin-based, and flatness-based. The main evaluation procedure of the paper follows that of Jiang et al. [23].

It's clear that the paper has contributions. However, I suggest the authors consider addressing the following issues after the paper is accepted.
- While the paper presents some interesting observational studies, two reviewers pointed out that the main conclusions lack insights. It is unclear how some measures correlate more with the robust generalization gap while others do not.
- It is also unclear if this paper presents a substantial conceptual improvement over the paper by Jiang et al. [23]. The paper seems to have followed mostly the evaluation procedures, as pointed out by one reviewer, and the authors have not discussed this point thoroughly in the paper.
- Other reviewers also provided useful feedback, such as on the title, writing clarity, and robustness of multiple attack methods (e.g., those used to avoid gradient obfuscation), including only those models exhibiting high training robustness.

I believe the authors have partially addressed some of their concerns during the rebuttal, but not all. I urge the authors to thoroughly address the remaining concerns in the final version.